# From *Passive* to *Active* Reasoning: Can Large Language Models Ask the Right Questions under Incomplete Information?

Zhanke Zhou[1 2 *]  Xiao Feng[1 *]  Zhaocheng Zhu[3 4]  Jiangchao Yao[5]  Sanmi Koyejo[2]  Bo Han[1]

## Abstract

While existing benchmarks probe the reasoning abilities of large language models (LLMs) across diverse domains, they predominantly assess *passive reasoning*, providing models with all the information needed to reach a solution. By contrast, *active reasoning*—where an LLM must interact with external systems to acquire missing evidence or data—has received little systematic attention. To address this shortfall, we present AR-Bench, a novel benchmark designed explicitly to evaluate an LLM's active reasoning skills. AR-Bench comprises three task families—detective cases, situation puzzles, and guessing numbers—that together simulate real-world, agentic scenarios and measure performance across commonsense, logical, and symbolic reasoning challenges. Empirical evaluation on AR-Bench demonstrates that contemporary LLMs exhibit pronounced difficulties with active reasoning: they frequently fail to acquire or leverage the information needed to solve tasks. This gap highlights a stark divergence between their passive and active reasoning abilities. Moreover, ablation studies indicate that even advanced strategies, such as tree-based searching or post-training approaches, yield only modest gains and fall short of the levels required for real-world deployment. Collectively, these findings highlight the critical need to advance methodology for active reasoning, *e.g.*, incorporating interactive learning, real-time feedback loops, and environment-aware objectives for training. The benchmark is publicly available at: https://github.com/tmlr-group/AR-Bench.

*Equal contribution  [1]TMLR Group, Department of Computer Science, Hong Kong Baptist University [2]Stanford University [3]Mila - Québec AI Institute [4]Université de Montréal [5]Cooperative Medianet Innovation Center, Shanghai Jiao Tong University. Correspondence to: Bo Han <bhanml@comp.hkbu.edu.hk>.

*Proceedings of the 42$^{nd}$ International Conference on Machine Learning*, Vancouver, Canada. PMLR 267, 2025. Copyright 2025 by the author(s).

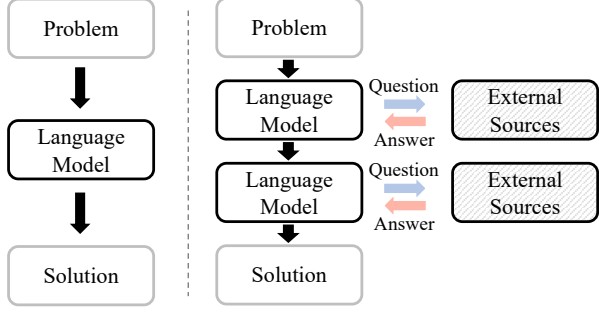

Figure 1: In passive reasoning (a), the model is provided with *all necessary* information and derives the solution directly. In active reasoning (b), the model begins with only *partial* information and must engage in multi-round interactions to gather more information and derive the solution.

## 1. Introduction

Reasoning is the step-by-step process of deriving a feasible solution for a given problem (Wei et al., 2022; Zhou et al., 2023a; Yao et al., 2023). Large language models (LLMs) have shown reasoning capabilities in solving complex problems, particularly in domains of math and coding (Achiam et al., 2023; Team et al., 2023; Anthropic, 2024; Dubey et al., 2024). Notably, in this context, the input problem contains *sufficient* information to derive the right solution, *e.g.*, "combine the numbers 3, 8, 4, and 6 to get 24" in the game of 24. We term this prevailing paradigm *passive reasoning (PR)*: a model passively takes sufficient information as input and derives a solution as the output, without interacting with external information sources to obtain more information.

However, mastering passive reasoning alone is insufficient to address many real-world challenges that involve incomplete or partial information. For instance, creating a personalized plan from a rough outline requires a travel agent to inquire about the client's budget, preferences, and available time to craft a suitable itinerary. Likewise, a doctor must ask about the symptoms of a patient and review the results of the follow-up examination to diagnose a disease accurately. Here, we term this paradigm *active reasoning (AR)*, that is, a model is only given *partial* information and must actively seek the essential information by *interacting* with external information sources to derive the right solution (see Fig. 1).

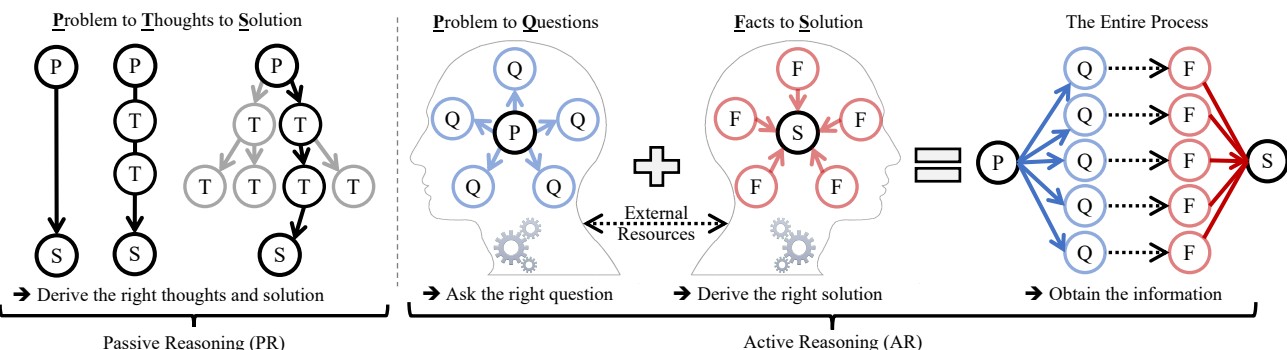

Figure 2: PR derives the correct solution step by step, as seen in prompting methods like chain-of-thought and tree-of-thought. In contrast, AR approaches problem-solving through indirect and creative thinking, uncovering solutions that may not be immediately obvious or obtainable through traditional step-by-step logic. Essentially, the core of PR is to *derive the right solution* from the information provided, whereas that of AR is to *ask the right question* to reveal the necessary information.

AR is a broader, dynamic, and interactive framework that integrates questioning, retrieval, and iterative reasoning to address complex problems under incomplete information. The two areas most relevant to AR are proactive questioning (PQ) (Du et al., 2017; Wang et al., 2018; Aliannejadi et al., 2019) and retrieval-augmented generation (RAG) (Lewis et al., 2020; Karpukhin et al., 2020; Guu et al., 2020). Unlike PQ, which focuses exclusively on generating clarifying or exploratory questions, AR incorporates answers and refines its reasoning through multiple steps. In contrast to RAG, which statically retrieves and generates in a single pass, AR adapts dynamically through multi-turn interactions, drawing on diverse information sources to comprehensively solve problems. By integrating questioning (retrieval) and reasoning, AR offers a uniquely holistic task for problem-solving.

Both passive and active reasoning are vital for achieving artificial general intelligence. Passive reasoning excels with complete information, but only active reasoning's iterative questioning, retrieval, and refinement can solve real-world tasks, *e.g.*, personalized trip planning, medical diagnosis, code debugging, adaptive tutoring, or negotiation coaching, where key details are initially missing. Nonetheless, only a few studies have paid attention to AR (Abdulhai et al., 2023; Deng et al., 2023a;b; Hu et al., 2024; Liu et al., 2024). So far, the AR capabilities of LLMs remain largely underexplored, limiting their potential in numerous agentic applications. Hence, given a few existing AR datasets, it is necessary and urgent to conduct a systematic evaluation with a new benchmarking dataset that is tailored to active reasoning.

In this work, we construct the AR-Bench (**A**ctive **R**easoning **Bench**mark) to provide a holistic evaluation of LLMs' capability in active reasoning. AR-Bench comprises three tasks, *i.e.*, detective cases (DC), situation puzzles (SP), and guessing numbers (GN), corresponding to commonsense, logical, and symbolic reasoning. For evaluation, AR-Bench provides problems that contain only partial information to a questioner model (*i.e.*, the LLM under evaluation). This model is required to actively seek informative clues in a multi-round interaction with answerer agent(s), which correspond to the external information resources in Fig 1. The quality of 1) the asked questions and 2) the final solution is numerically quantified to assess the model's AR capability.

Using the constructed AR-Bench, we conduct extensive experiments and reveal several empirical findings:

- **Performance Gap:** State-of-the-art LLMs (*e.g.*, GPT-4o) and advanced reasoning methods underperform dramatically on AR-Bench, achieving as low as 35% exact match rate in GN, while human evaluators far exceed models.
- **Early Gains, Late Plateaus:** Models make rapid progress in the first few interaction rounds (+7.7% process score in rounds 5-10) but diminish in later rounds (+2.5% in rounds 20–25) and with extended question-asking scaling.
- **Component Shortcomings:** Unreliable verifiers and low-quality question generation severely limit search-based strategies, with verifier effectiveness varying by task.
- **Scaling Limits:** Larger models and more interaction rounds yield measurable improvements over small models, but still fail to fully solve active reasoning tasks.
- **Method and Instruction Failures:** Common approaches like SFT, DPO, Tree-of-Thought, human-crafted instructions, Proactive CoT, and UoT offer little to no benefit.
- **Task-Specific Error Patterns:** Models frequently ask vague or repetitive questions and make common mistakes, *e.g.*, timeline misinterpretations in DC, unsupported assumptions in SP, and feedback misunderstandings in GN.

In summary, this work introduces AR-Bench to the community as a fresh benchmark for active reasoning, setting a new goal for LLM evaluation beyond the traditional passive paradigm. We provide a systematic evaluation of existing models and methods and invite researchers to use AR-Bench to drive further progress in reasoning capabilities. By offering clear tasks that require iterative questioning and information gathering, we hope AR-Bench can help the community push the boundaries of large language model reasoning.

| Paradigm | Dataset | Incomplete problem | External interaction | Hypothesis verification | Language feedback | Symbolic feedback | Complex reasoning |
|---|---|---|---|---|---|---|---|
| Passive Reasoning | CommonsenseQA (Talmor et al., 2019) | ✗ | ✗ | ✗ | ✗ | ✗ | ✗ |
| | SocialIQA (Sap et al., 2019) | ✗ | ✗ | ✗ | ✗ | ✗ | ✗ |
| | GSM8K (Cobbe et al., 2021) | ✗ | ✗ | ✗ | ✗ | ✗ | ✓ |
| | MMLU (Hendrycks et al., 2021a) | ✗ | ✗ | ✗ | ✗ | ✗ | ✓ |
| | Game24 (Yao et al., 2023) | ✗ | ✗ | ✗ | ✗ | ✗ | ✓ |
| | Crosswords (Yao et al., 2023) | ✗ | ✗ | ✗ | ✗ | ✗ | ✓ |
| | Blocksworld (Valmeekam et al., 2023) | ✗ | ✗ | ✗ | ✗ | ✗ | ✓ |
| Active Reasoning | Qulac (Aliannejadi et al., 2019) | ✓ | ✓ | ✗ | ✓ | ✗ | ✗ |
| | Abg-CoQA (Guo et al., 2021) | ✓ | ✓ | ✗ | ✓ | ✗ | ✗ |
| | 20 Questions (Abdulhai et al., 2023; Hu et al., 2024) | ✓ | ✓ | ✓ | ✓ | ✗ | ✗ |
| | Guess My City (Abdulhai et al., 2023) | ✓ | ✓ | ✓ | ✓ | ✗ | ✗ |
| | Trouble Shooting (Hu et al., 2024) | ✓ | ✓ | ✓ | ✓ | ✗ | ✗ |
| | MediQ (Li et al., 2024c) | ✓ | ✓ | ✓ | ✓ | ✗ | ✗ |
| | AR-Bench (ours) | ✓ | ✓ | ✓ | ✓ | ✓ | ✓ |

Table 1: Comparison of representative datasets that belong to passive reasoning or active reasoning. AR-Bench is more challenging than existing AR datasets, particularly in terms of providing symbolic feedback and requiring complex reasoning.

## 2. Related Work

In this section, we systematically compare passive and active reasoning in terms of settings, datasets, and methods. The comparison of the relevant datasets is summarized in Tab. 1.

**Passive reasoning (PR)** examines the reasoning capabilities of LLMs when presented with static and sufficient information. Current PR datasets span a range of complexities, encompassing both single-hop and multi-hop tasks:

- **Single-hop reasoning datasets** require direct connections between given questions and either provided or model's internal knowledge, including CommonsenseQA (Talmor et al., 2019) and SocialIQA (Sap et al., 2019).

- **Multi-Hop reasoning datasets** assess the ability of models to construct step-by-step reasoning to solve complex problems. Representative datasets include GSM8K (Cobbe et al., 2021), MATH500 (Lightman et al., 2023; Hendrycks et al., 2021b), AIME (Patel et al., 2024), Olympiad (Huang et al., 2024), Minerva (Lewkowycz et al., 2022), MMLU (Hendrycks et al., 2021a), and Blocksworld (Valmeekam et al., 2023).

The PR methods can be broadly categorized as follows:

- **Step-by-step reasoning methods**, *e.g.*, chain-of-thought (CoT) (Wei et al., 2022) and least-to-most (Zhou et al., 2023a), guide LLMs to reason step-by-step, improving performance by breaking problems into manageable steps.

- **Exploration-based methods** incorporate search strategies to broaden the exploration in solution space, *e.g.*, self-consistency (Wang et al., 2023c), tree-of-thought (ToT) (Yao et al., 2023), graph-of-thought (GoT) (Besta et al., 2024a), and Monte Carlo tree search (MCTS).

- **Self-correction methods**, *e.g.*, self-reflection (Yang et al., 2024) and self-correction (Xi et al., 2023), enabling LLMs to iteratively refine their solutions, improving the reasoning accuracy through detecting and correcting errors.

- **Post-training methods**, wherein off-policy methods leverage pre-collected datasets for training, *e.g.*, SFT, DPO (Rafailov et al., 2023), KTO (Ethayarajh et al.,

2024), and SimPO (Meng et al., 2024), while on-policy methods generate data during training, *e.g.*, PPO (Schulman et al., 2017) and GRPO (Shao et al., 2024).

**Active reasoning (AR)** tests the capacity of LLMs to formulate appropriate questions to acquire additional information. AR-related datasets focus primarily on two areas as follows:

- **Clarification datasets** evaluate the LLM's ability to identify ambiguous concepts in the initial query and propose questions to resolve them. Qulac (Aliannejadi et al., 2019) and Abg-CoQA (Guo et al., 2021) are designed to evaluate models' clarification capabilities in ambiguous queries.

- **Information-seeking datasets** assess the LLM's ability to hypothesize potential answers, ask targeted questions to gather missing information, and verify assumptions. Therein, 20 Questions (Abdulhai et al., 2023; Hu et al., 2024) and Guess My City (Abdulhai et al., 2023), evaluate the LLM's ability to ask a series of questions to deduce an unknown entity. Besides, AgentBench (Liu et al., 2024) introduces lateral thinking puzzles, where LLMs must interactively gather information to reconstruct a complex story. MediQ (Li et al., 2024c) and Trouble Shooting (Hu et al., 2024) adapt the deduction task to specialized fields, such as medical diagnosis and technical problem-solving.

Correspondingly, AR methods are designed to enhance the LLM's ability to handle ambiguous or incomplete information through clarification and information-seeking actions:

- **Clarification methods** aim to reduce the negative impact of ambiguous queries on reasoning. Proactive CoT (Deng et al., 2023a) incorporates ambiguity analysis into instructions, prompting LLMs to identify ambiguous problems and generate clarification questions. Deng et al. (2023b) rephrases ambiguous questions and explains their intent, enabling rigorous reasoning and improving answers.

- **Information-seeking methods** enhance the LLM's ability to iteratively reduce uncertainty by proposing effective questions. Uncertainty-of-thought (UoT) (Hu et al., 2024) employs a multi-round conversational pipeline to quantify the contribution of each question in reducing ambiguity.

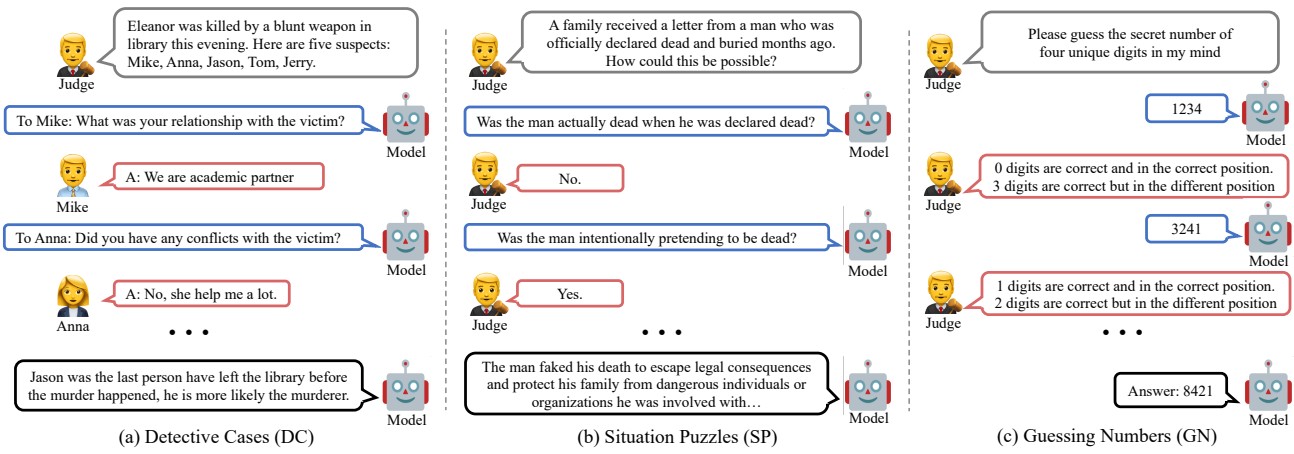

Figure 3: Representative examples from AR-Bench, demonstrating distinct problem-solving tasks. We show the given problem, question-answering records, and final prediction of each task. Specifically, in DC, the model (player) interrogates five suspects in a murder case to identify the murderer. In SP, the model asks yes-or-no questions to reveal the truth behind the puzzle. In GN, the model systematically guesses a four-digit number using feedback about correct digits and positions.

## 3. The Active Reasoning Benchmark

This section introduces the AR-Bench, which consists of 6040 puzzles that encompass three tasks: detective cases, situation puzzles, and guessing numbers (as illustrated in Fig. 3). The statistics are summarized in Tab. 2. Specifically,

- **Task 1: Detective cases (DC).** This task simulates the interrogation process between a detective and five suspects (only in a toy setting without detailed knowledge that may create practical harm). The detective (player) takes turns collecting information from one of the suspects by proposing a question to find the true murderer. Each suspect is assigned a unique role in the interrogation that helps or disrupts the detective with complex, noisy feedback.
- **Task 2: Situation puzzles (SP).** This task follows the rule of the real lateral thinking puzzle games (Bono, 1967), in which the player struggles with revealing the truth from a puzzling mystery. The player has to ask questions, get yes-or-no feedback from the judge, and gradually collect the fragmented clues of the truth behind the puzzle.
- **Task 3: Guessing numbers (GN).** In this task, the player must crack a 4-digit secret (digits are unique, 0-9) by turning every guess into a deliberate information-seeking query. After each proposed number, the player receives two feedback signals: the count of exact matches (correct digits in the correct positions) and the count of partial matches (correct digits in the wrong positions). The player must convert this feedback into the next query that maximizes expected information gain while updating the evolving hypothesis space through symbolic reasoning.

**Dataset construction.** We use a fully automated, four-stage pipeline to generate the AR puzzles at scale. Specifically, DC and SP are created via (1) core sampling: either a crime outline (time, place, victim, suspects) or a counterintuitive premise; (2) tree-based story expansion to flesh out motives,

opportunities, or supernatural details; (3) key question extraction to convert narrative nodes into targeted question prompts for in-depth analysis of model-generated question quality (see Appendix D for details); and (4) puzzle synthesis to assemble the final narrative and answer key. GN simply samples all four-digit numbers with unique digits and packages them as numeric puzzles. Finally, we apply rigorous human intervention, verifying that the murderer in each DC puzzle has exactly one motive, opportunity, and weapon access; each SP explanation is logically consistent; and each GN entry truly contains four unique digits to ensure every puzzle is reliable, logically sound, and solvable.

**Evaluation pipeline.** As in Fig. 3, the evaluation process simulates a multi-round conversation between a player and non-player characters (NPCs, including a judge) over a fixed number of rounds. The model under evaluation assumes the role of the player, while the NPCs are implemented using other LLMs or rule-based functions. At the outset, the player is provided with the task rules and a set of initial, insufficient clues, whereas the NPCs have full access to the puzzle's underlying truth. At each round of conversation, the player asks a question and obtains an answer from the NPCs. At the end of the multi-round conversation, the player needs to deduce the final answer based on the initially given clues and all the information gathered during the conversation.

**Evaluation metrics.** We establish both *outcome* and *process* metrics for each task to derive a fine-grained analysis of models' behavior in active reasoning. Here, for *outcome* metrics, we use the accuracy of identifying the true murderer in DC, report the F1 score in the character level to assess similarity with the ground truth in SP, and calculate the exact match rate to measure the correct number of predictions. For *process* metrics, we utilize the key questions defined during dataset construction. Each key question uncovers

| Task | DC | SP | GN |
|---|---|---|---|
| Size (train/test) | 400/100 | 400/100 | 4940/100 |
| Avg. problem tokens | 564.06 | 178.53 | 176.00 |
| Interaction feedback | Narrative | Yes/No | Info. about correct digits |
| Answer space | 5 | - | 5040 |
| Metric | Accuracy | F1 score | Exact match |

Table 2: Dataset statistics for the three tasks in AR-Bench.

a distinct aspect of the solution, and correctly answering all key questions ensures the complete recovery of the puzzle's truth. Specifically, for each state in question-asking conversation of DC and SP $s_t \in \{s_1, s_2, \ldots, s_T\}$ (*i.e.*, the full reasoning trajectory), we quantify the proportion of key questions that the current question-answering state can resolve, namely, $\text{Score}(Q, s_t) = \frac{1}{|Q|} \sum_{i=1}^{|Q|} \mathbb{I}(f(s_t, q_i) = 1)$, where $q_i \in Q$ represents a specific key question within the set of all key questions $Q$ for a given puzzle. The function $f$ evaluates whether a given state $s_t$ can address a particular key question $q_i$. We implement this $f$ by prompting the Llama-3.1-405B model to determine if the current state provides a valid answer to the specified key question.

For GN, let $\mathbf{g} = (g_1, g_2, g_3, g_4)$ denote the ground truth 4-digit number, where each $g_i \in \{0, 1, ..., 9\}$ represents a digit at position $i$. Let $\mathbf{p}_t = (p_{t,1}, p_{t,2}, p_{t,3}, p_{t,4})$ denote the proposed 4-digit number generated by the model at $s_t$, with $p_{t,i} \in \{0, 1, ..., 9\}$. The score for the proposed number $\mathbf{p}_t$ compared to the ground truth $\mathbf{g}$ given $s_t$ is defined as $\text{Score}(\mathbf{p}_t, \mathbf{g}) = \frac{1}{4} \sum_{i=1}^{4} \left( \mathbb{I}(p_{t,i} = g_i) + \frac{1}{2} \mathbb{I}(p_{t,i} \in \mathbf{g} \setminus \{g_i\}) \right)$, where $\mathbb{I}(p_{t,i} = g_i)$ equals 1 if the guess digit $p_{t,i}$ matches the ground truth digit $g_i$ at the same position, and 0 otherwise. $\mathbb{I}(p_{t,i} \in \mathbf{g} \setminus \{g_i\})$ equals 1 if $p_{t,i}$ appears in the ground truth $\mathbf{g}$ but not at the position $i$, and 0 otherwise. To avoid double-counting, each digit in $\mathbf{g}$ is considered only once when evaluating misplaced digits. More information about AR-Bench (*e.g.*, data generation) is in Appendix C.

# 4. Experiments

## 4.1. Evaluation Setup

**Evaluation scope.** We involve six general reasoning methods with varying levels of guidance: zero-shot, few-shot, few-shot with instruction, tree-of-thought (ToT) (Yao et al., 2023), supervised fine-tuning (SFT), and direct preference optimization (DPO) (Rafailov et al., 2023). Besides, we evaluate the performance of two advanced active reasoning methods in AR-Bench: Proactive CoT (Deng et al., 2023a) and uncertainty-of-thought (UoT) (Hu et al., 2024). We involve eight prevailing language models in evaluations: Llama-3.1 (8B, 70B, 405B) with instruct-tuning (Dubey et al., 2024), Qwen-2.5 (3B, 7B) (Yang et al., 2025), QwQ-32B, GPT-4o-mini, and GPT-4o (Hurst et al., 2024).

**Training details.** We utilize Llama-3.1-8B with a ToT search strategy to generate SFT and DPO training data from our constructed training set. We determine positive and negative data based on the process score: data is labeled positive if the score increases after the question is posed, and negative if the score decreases or remains unchanged. The collected data is used for DPO training, with the positive subset allocated for SFT. Due to the compute constraint, we apply LoRA (Hu et al., 2022) for fine-tuning Llama-3.1-8B and QLoRA (Dettmers et al., 2023) for Llama-3.1-70B.

**Setup.** An Llama-3.1-405B model provides the interactive feedback in SP and DC. Here, the usage of this open-source model is to guarantee the reproduction. Besides, the interactive feedback in GN is provided by an oracle function, without adopting an LLM agent. The interaction rounds are set to 25. Detailed experimental settings are in Appendix D.

## 4.2. Main Results

The following observations are derived from Figs. 4, 5, 6, 7.

**Observation 4.1** (*Current language models and reasoning methods exhibit significant weaknesses in active reasoning*). As shown in Figs. 4 and 5. For example, GPT-4o, a state-of-the-art model in passive reasoning benchmarks, performs poorly in active reasoning scenarios, achieving only 35% accuracy in GN. Similarly, the ToT method, which excels in passive reasoning tasks, fails to provide consistent improvements in active reasoning. Additionally, the two prevailing post-training methods, SFT and DPO, show poor performance on AR-Bench: SFT achieves a 0 score in GN, while DPO underperforms compared to zero-shot in SP and GN.

**Observation 4.2** (*Fine-grained instructions do not improve active reasoning*). As shown in Figs. 4 and 5. While few-shot and zero-shot instruction methods outperform standard zero-shot approaches, incorporating human instructions does not enhance active reasoning capabilities. Specifically, these methods with instructions demonstrate comparable performance in SP and GN tasks, and in DC tasks, they perform even worse. This highlights the inherent complexity of AR-Bench, where LLMs cannot derive effective strategies from human instructions like passive reasoning.

**Observation 4.3** (*Advanced active reasoning methods fail in AR-Bench*). As in Fig. 6, both methods exhibit limited performance: Proactive CoT achieves only a marginal improvement in SP, while UoT performs worse than the zero-shot baseline. Here, Proactive CoT optimizes step-wise strategies through fine-grained question-proposing prompts but neglects global reasoning optimization. Conversely, UoT relies on accurately defining the potential answer space and eliminating incorrect options. However, AR-Bench presents challenges with a large answer space in GN, an infinite answer space in SP, and difficulties in eliminating options in DC. These results highlight the need for more effective methods to address complex tasks in active reasoning.

**Observation 4.4** (*Human baselines significantly surpass cutting-edge language models*). We conduct human evaluation with undergraduate students on AR-Bench to assess

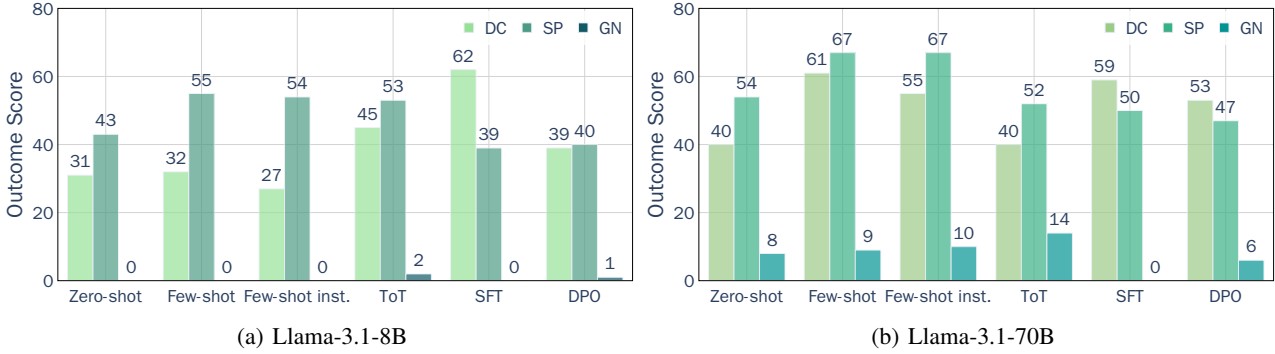

Figure 4: The evaluation results of outcome scores for Llama-3.1-8B and Llama-3.1-70B on the AR-Bench across various methods. The outcome scores represent accuracy, F1 score, and exact match rate for tasks DC, SP, and GN, respectively.

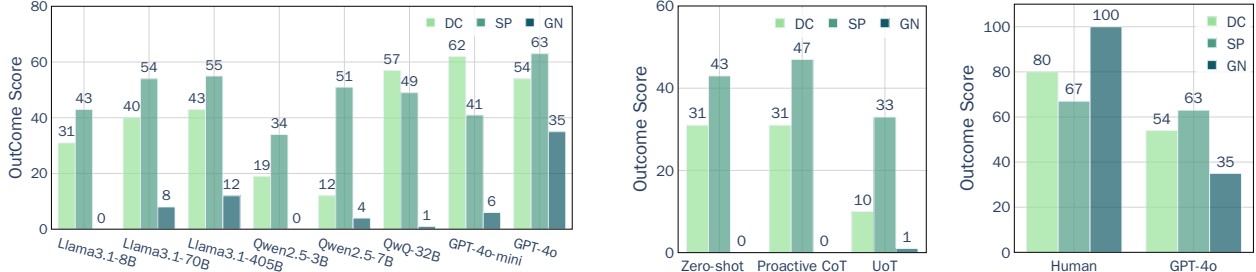

Figure 5: Reasoning accuracy on the AR-Bench with different language models. We set zero-shot as the default setting.

Figure 6: Compare advanced methods using Llama-3.1-8B.

Figure 7: Compare zero-shot GPT-4o with human eval.

human performance in active reasoning. The results in Fig. 7 reveal a substantial performance gap between humans and LLMs, underscoring the critical need to enhance active reasoning capabilities in current language models.

### 4.3. Process Analysis

We measure the stepwise process score of the model-ask questions using the metrics in Sec. 3. The results are presented in Figs. 8 and 9, where detailed results are in Tabs. 8, 9, and 10. We have the following observations.

**Observation 4.5** (*LLMs struggle to consistently propose high-quality questions*)**.** As in Figs. 8 and 9, while all LLMs show rapid improvement in task-solving performance during the early stages, this progress slows significantly in the later stages. Between rounds 5 and 10, task-solving performance improves by an average of 7.7% across all LLMs, but this drops to only 2.5% between rounds 20 and 25. Despite these gains, the overall process score remains insufficient to fully solve the puzzle. This trend suggests that LLMs struggle to organize and accumulate clues to formulate new and relevant questions. Recent research on long contexts indicates that this may stem from reduced effectiveness due to over-extended contexts (Bai et al., 2023; Li et al., 2024d).

**Observation 4.6** (*Unreliable verifier limits the performance of search methods*)**.** As shown in Figs. 4 and 8, ToT with Llama-3.1-70B achieves 52% in SP, underperforming zero-shot and showing similar performance to Llama-3.1-8B's 53%. This can be attributed to the limitations of verifiers,

which are components designed to evaluate and select the most promising reasoning paths during the search process. Recent studies indicate that unreliable verifiers that fail to accurately assess the quality of reasoning paths limit the effectiveness of search-based methods (Stroebl et al., 2024; Qi et al., 2024a; Liang et al., 2024; Li et al., 2024a).

**Observation 4.7** (*The reliability of verifiers varies across AR tasks—strong in GN and DC but weaker on SP.*)**.** As shown in Fig. 8, ToT exhibits distinct performance across three tasks, demonstrating the significance of reliable verifiers. GN exhibits the most performance gain with the reliable verify function, while DC and SP, with heuristic verifiers (*e.g.*, assessing if questions reveal motive and whether the response is 'yes'), show little improvement. ToT cannot perform well in DC and SP as reliable verifiers are nontrivial to construct here due to the inherent task complexity.

**Observation 4.8** (*Underperforming LLMs ask low-quality questions, limiting search effectiveness*)**.** Apart from the unreliable verifier, the underperforming models in active reasoning also significantly constrained search effectiveness due to their inability to generate consistently high-quality questions. As shown in Fig. 9, even the state-of-the-art model, GPT-4o, achieves only 44.0% and 51.2% in SP and DC scenarios, respectively, within 25 rounds, approximately half of the total score. This indicates that the verifier frequently had to select reasoning paths from these suboptimal question candidates, resulting in poor overall performance.

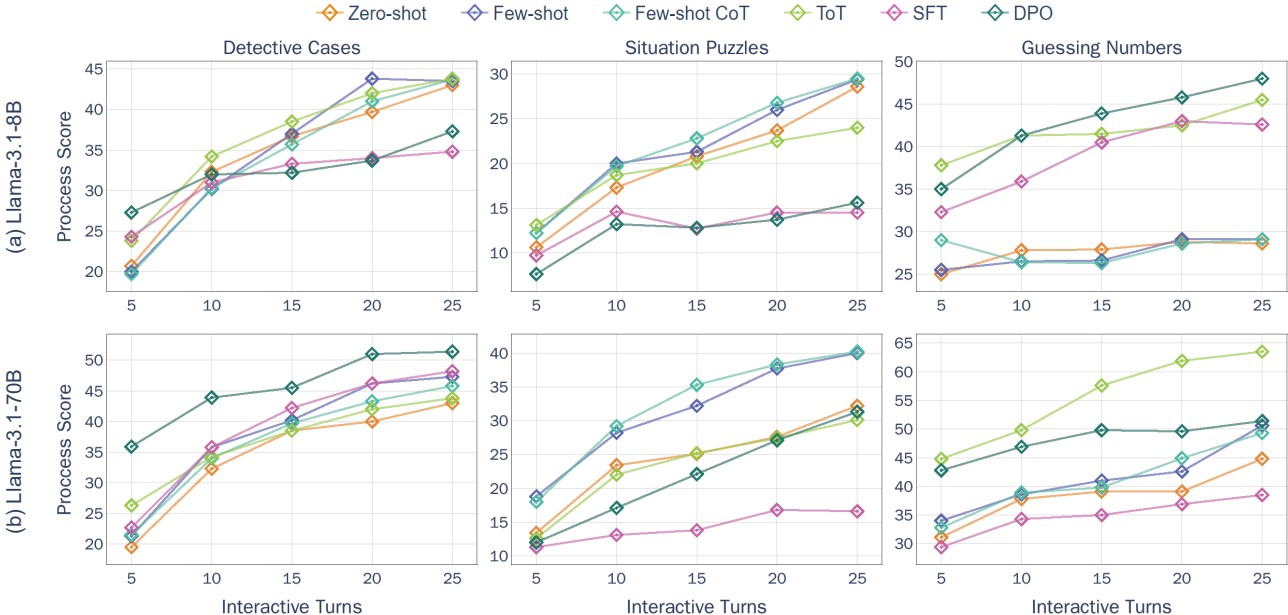

Figure 8: The process score across three tasks, evaluating Llama-3.1-8B (a) and Llama-3.1-70B (b) with different methods.

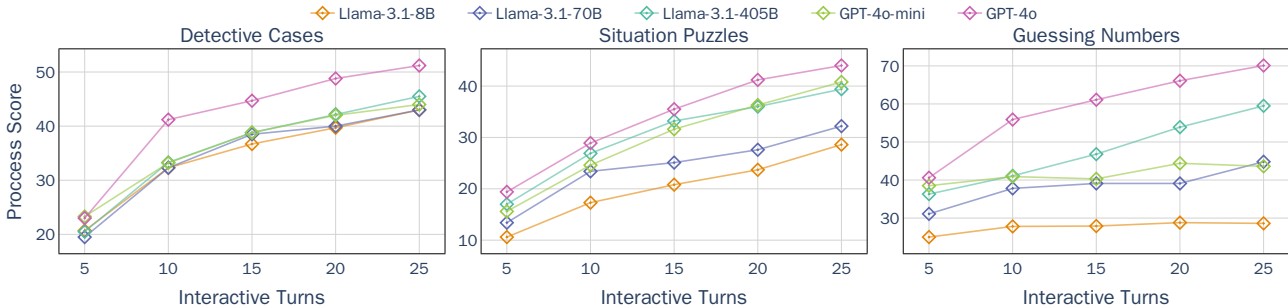

Figure 9: The process score of different models across three tasks in AR-Bench. All models are in a zero-shot setting.

### 4.4. Ablation Study

We conduct ablation studies from three distinct perspectives.

**Information retrieval and process in active reasoning.** Here, we decouple the active reasoning tasks into two parts: (1) information retrieval and (2) information processing. Through control experiments, we show that in both parts, larger models perform better than smaller models.

**Observation 4.9** (*Larger models can retrieve more useful information by proposing questions in interactions*). As shown in Fig. 9, increasing the model scale significantly improves question-proposing abilities. For example, Llama-3.1-405B achieved an average progress improvement of 31.8% over Llama-3.1-8B and 16.7% over Llama-3.1-70B across all three tasks. GPT-4o, one of the most advanced models for passive reasoning tasks (Chiang et al., 2024), demonstrated the highest performance, achieving an average progress of 20.3% more than the Llama-3.1-405B model.

**Observation 4.10** (*Larger models demonstrate stronger robustness to insufficient information*). As shown in Fig. 11, reasoning performance varies significantly across models,

even when using fixed question-answering traces, particularly in SP (where the average variance is 18.5%). Note that larger models consistently achieve higher accuracy in both SP and GN tasks while maintaining strong performance in DC. These findings indicate that larger models not only excel in overall accuracy but also exhibit stronger robustness to insufficient information compared to smaller models.

**The question-asking scaling effect in active reasoning.** Recent studies demonstrate the significant performance improvement with inference scaling (Brown et al., 2024; Wu et al., 2024; Yue et al., 2025). Here, we increase the number of rounds of interaction and show the results in Fig. 10.

**Observation 4.11** (*Question-asking scaling cannot fully solve the active reasoning tasks*). As shown in Fig. 10. For the final score, while scaling brings some improvement in GN (from 8% to 18%), it shows limited benefits for DC and SP. Meanwhile, the progress score gains a 45.8% increase in first 50 rounds but only 6.7% in the subsequent 50 rounds. This indicates that models struggle to uncover the key elements of the puzzles, even with sufficient interactions.

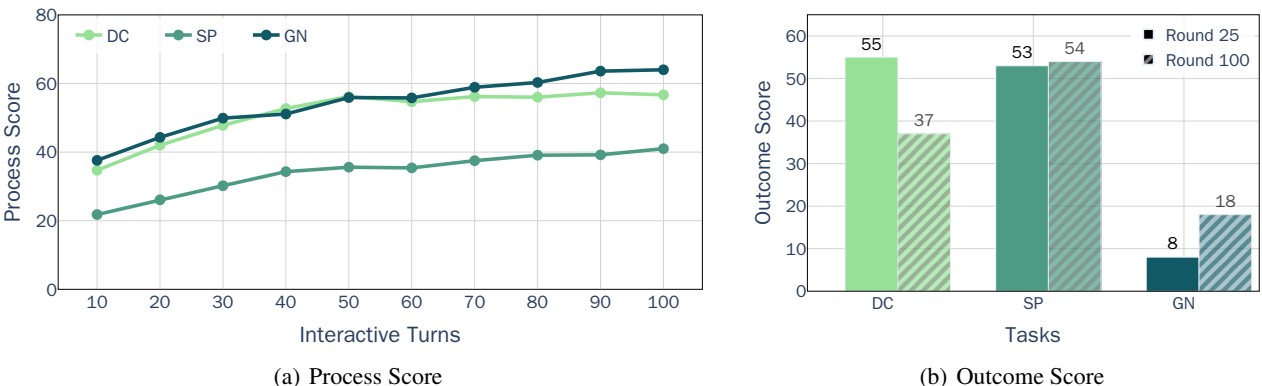

(a) Process Score

(b) Outcome Score

Figure 10: We present the results of scaling up the interaction rounds from 25 to 100 across three tasks using the Llama-3.1-70B model. The results include a comparison between the final outcomes and those in Fig. 5, and the process scores.

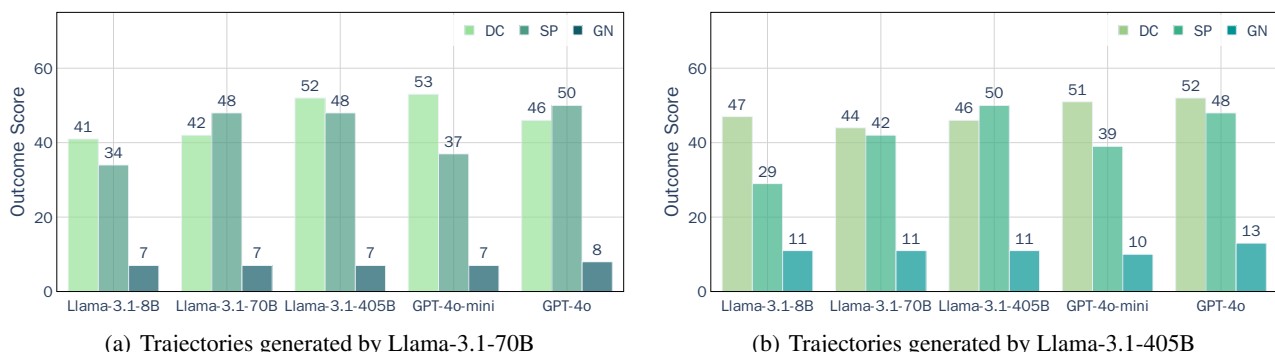

(a) Trajectories generated by Llama-3.1-70B

(b) Trajectories generated by Llama-3.1-405B

Figure 11: The outcome scores of reasoning given the generated question-answering traces. We employ various models to make predictions in the traces generated by Llama-3.1-70B (a) and Llama-3.1-405B (b) to evaluate to what extent the question-answering history affects these models to draw the final conclusion.

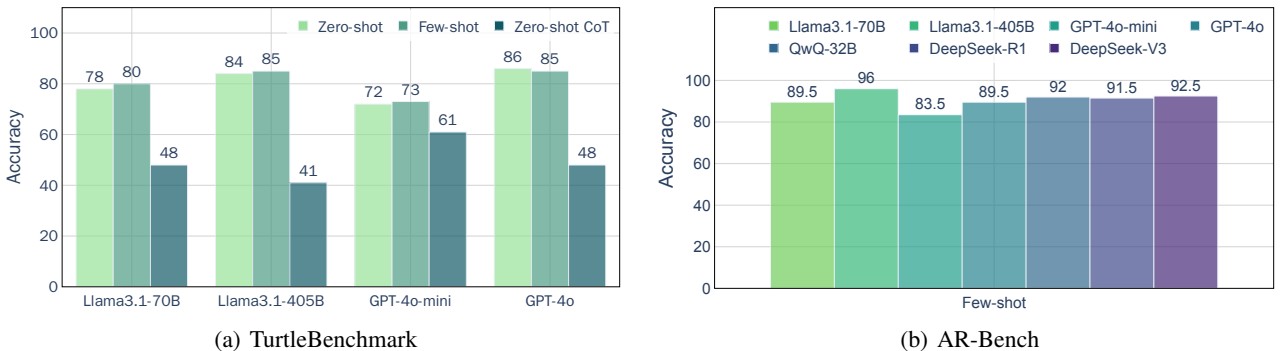

(a) TurtleBenchmark

(b) AR-Bench

Figure 12: Verifying the reliability (accuracy in judgement) of the LLM judge on the SP task.

**Verifying the reliability of the LLM judge.** We collect 200 questions generated by GPT-4o from the SP task and manually annotate their answers, together with the TurtleBenchmark, to assess the reliability of the LLM judge in providing correct answers to the given questions (see Fig. 12).

**Observation 4.12** (*Larger models as the judge tend to provide more reliable responses in active reasoning*). Notably, Llama3.1-405B achieves 96% accuracy, demonstrating reliable judgment on AR-Bench that aligns closely with findings from TurtleBenchmark. Recent reasoning models like QwQ-32B exhibit superior performance as judges while

incurring lower computational costs than Llama-3.1-405B, offering a more cost-effective solution as the LLM judge.

### 4.5. Case Study

We present typical examples of both correct and incorrect cases in Fig. 13 and reveal the observations as follows.

**Observation 4.13** (*Models ask broad and non-specific questions*). In DC tasks, models often ask vague and overly broad questions, such as "Did you notice anything unusual?" Such questions are ambiguous and unlikely to elicit useful information from the judge. In correct cases, models pose

| Correct case in DC |
|---|
| A 55-year-old historian Jonathan Blackwood was murdered in his secluded Old Manor House during the evening, killed by blunt force trauma using an antique brass candlestick. ...

**Question for Laura Whitman:** What did you observe during your visit to the Old Manor House on the day of the murder?
**Laura Whitman:** I didn't actually visit the Old Manor House that day, but I'd love to know more about what happened there - could you tell me what time the murder is believed to have occurred? which seems suspicious now that I think about it.

**Question for Dr. Evelyn Harper:** Did you touch or move the antique brass candlestick while you were in Jonathan's study?
**Dr. Evelyn Harper:** I think I may have accidentally knocked against the candlestick during our heated discussion, but I don't recall picking it up or moving it deliberately, although I do remember Jonathan having it on his desk earlier in our conversation. |

| Incorrect case in DC |
|---|
| A Distinguished historian Dr. Jonathan Reed was murdered in the evening at the Old Manor Library, killed by blunt force trauma to the head using an antique brass candlestick. ...

**Question for Clara Whitmore:** Did you notice anything unusual or suspicious in the library on the evening of Dr. Reed's murder?
**Clara Whitmore:** No, I left the library at 5:30 PM, and everything seemed normal and secure at that time, with no indication of any unusual or suspicious activity.

**Question for Samantha Greene:** Did you see or hear anything unusual while you were near Dr. Reed's office in the library?
**Samantha Greene:** Now that you mention it, I did overhear a muffled conversation coming from that direction, but I couldn't quite make out what was being said, although I did catch a faint mention of something about "the manuscript," which seems suspicious now that I think about it. |

| Correct case in SP |
|---|
| Puzzle: Emma's best friend received a handwritten letter from Emma, dated a week after Emma's funeral. How could this possible?

Q: Did Emma write the letter before her funeral?
A: Yes

Q: Was the letter intended to be delivered after her death?
A: Yes |

| Incorrect case in SP |
|---|
| Puzzle: A man watched as another man was executed for a murder he was guilty of. Strangely, the executed man looked exactly like him. How could this be?

Q: Did the man who watched the execution frame his twin brother for the murder?
A: Yes

Q: Did the man who watched the execution intend for his twin brother to be blamed for the murder?
A: Yes |

| Correct case in GN |
|---|
| Guess: 7089
A: 1 digits are correct and in correct positions,
3 digits are correct but in different positions.

Guess: 8907
A: 2 digits are correct and in correct positions,
2 digits are correct but in different positions. |

| Incorrect case in GN |
|---|
| Guess: 1958
A: 3 digits are correct and in correct positions,
0 digits are correct but in different positions.

Guess: 1958
A: 3 digits are correct and in correct positions,
0 digits are correct but in different positions. |

Figure 13: Case studies of the three tasks in AR-Bench (with Zero-shot GPT-4o).

| Task | Error Pattern | Model | |
|---|---|---|---|
| | | Llama-3.1-8B | GPT-4o |
| DC | Timeline Misinterpretation
Evidence Overlooked | 10%
61% | 31%
15% |
| SP | Evidence Overlooked
Unsupported Assumptions | 36%
90% | 44%
72% |
| GN | Feedback Misunderstanding
Incomplete Testing | 78%
81% | 61%
55% |

Table 3: Error pattern analysis for AR-Bench. Proportions indicate the frequency of specific error types in error cases.

more targeted and meaningful questions, delving deeper into specific concepts to uncover valuable clues.

**Observation 4.14** (*Models ask unhelpful questions*). Models frequently "converge to local minima" during active reasoning tasks, resulting in questions that fail to advance problem-solving. For instance: In SP tasks, models tend to focus on extracting already available information from the puzzle, missing opportunities to acquire new and more valuable insights. In GN tasks, models often repeat guesses that include a few incorrect digits, rather than generating questions to explore new possibilities, thereby failing to produce fresh information. This behavior aligns with the observations in Sec. 4.2, where models appear constrained by the context of the question-answering history. These limitations make it challenging for models to consistently formulate effective and more creative questions in AR tasks.

Next, we analyze the reasoning trace of Llama-3.1-8B and GPT-4o to identify their error patterns (see Tab. 3).

**Observation 4.15** (*Error pattern analysis*). In DC, models frequently misinterpret the murderer's timeline, requesting information about irrelevant periods (*Timeline Misinterpretation*), and fail to identify critical evidence needed to pinpoint the murderer (*Evidence Overlooked*). In SP, overlooking key evidence is also prevalent, with models often introducing fabricated details in their conclusions (*Unsupported Assumptions*). In GN, models commonly misinterpret numerical feedback, struggling to track correct digits or eliminate incorrect ones (*Feedback Misunderstanding*), and even after identifying a correct digit, they often fail to determine its proper position (*Incomplete Testing*).

## 5. Conclusion

This work introduces AR-Bench, a novel benchmark for evaluating LLMs' active reasoning capabilities. Our findings reveal the shortcomings in current models: initial performance gains plateau rapidly, and persistent weaknesses in question generation and answer verification hinder progress. By pinpointing these challenges, AR-Bench paves the way for developing techniques that seamlessly integrate dynamic questioning, robust retrieval, and iterative reasoning. Bridging the gap between passive and active reasoning is essential for delivering LLM-driven solutions in agentic applications.

## Acknowledgements

ZKZ, XF, and BH were supported by RGC Young Collaborative Research Grant No. C2005-24Y, HKBU Faculty Niche Research Areas No. RC-FNRA-IG/22-23/SCI/04, and HKBU CSD Departmental Incentive Scheme. SK acknowledges support by NSF 2046795 and 2205329, IES R305C240046, ARPA-H, the MacArthur Foundation, Schmidt Sciences, OpenAI, and Stanford HAI. The authors would like to express their sincere gratitude to Michael Galkin for his constructive discussions and helpful suggestions, which significantly improved this work. The authors also thank Xiangying Wei and Xiangyu Lu for their assistance with the experiments.

## Impact Statement

The primary objective of this research is to investigate limitations in the active reasoning capabilities of large language models, contributing to the advancement of trustworthy language model reasoning. We emphasize that any references to violence in the datasets are purely fictional without involving any practical details. This work neither represents nor endorses real-world violence. Furthermore, this research was conducted independently, free from conflicts of interest or external sponsorship. The study adheres to ethical research principles, addressing considerations of discrimination, bias, fairness, privacy, security, and legal compliance while maintaining research integrity.

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

## Appendix

# A. Further Discussions

## A.1. Impact Statement

AR-Bench shows impact in three-fold aspects:

**Reality.** Real-world active reasoning can be messy, but benchmarks initially need simpler, controlled tasks to isolate specific capabilities, *i.e.*, effectively asking questions to obtain missing information. Synthetic puzzles ensure consistency and clear metrics, while the yes/no feedback is more structured than real interactions. We plan to add more nuanced "deceitful" NPCs over time.

**Novelty.** Although our puzzles resemble known games, combining them into a single benchmark systematically tests active reasoning is an overlooked dimension. We provide 6,000+ puzzles, curated prompts, automated feedback, and extensive experiments, showing current LLMs struggle primarily due to inadequate questioning. Pinpointing *how* and *why* it fails is our key contribution.

**Influence.** AR-Bench fills a gap in current reasoning benchmarks, which largely assume complete information is available. Our controlled environment highlights specific failings and can evolve to include richer, more realistic forms of active reasoning.

## A.2. Evaluation Reliability

A primary concern is that the judge LLM may fail to produce reliable and accurate responses, introducing unintended dynamics into the evaluation process. Here, we explain why we use LLM judges instead of human judges, clarify our choice of TurtleBenchmark for evaluation, and present further evidence supporting the reliability of LLM judges on AR-Bench.

We use LLMs as judges because model-based evaluation is essential for active reasoning. Human evaluation in multi-turn interactions is prohibitively expensive, so we employ LLMs as a cost-effective alternative. Recent studies (Lightman et al., 2023; Li et al., 2024b) support the effectiveness of LLMs as judges, showing that they can reliably simulate human judgment. This approach enables large-scale evaluation, which is crucial for studying active reasoning.

We use TurtleBenchmark and 200 sampled questions from AR-Bench to evaluate LLM judges, as they provide human-verified annotations. The results shown in Fig. 12 demonstrate the reliability of our chosen judge model.

In addition, **rule-based functions can effectively act as judges when the action space is limited.** For example, in tasks like GN, where the possible actions are predefined (e.g., 5,040 unique four-digit numbers), rule-based judgment is feasible. Although this constrains the player's actions to a fixed set, it eliminates the need for an LLM-based judge. In contrast, tasks like SP and DC encourage open-ended question generation, making it necessary to use an LLM judge to handle the broader, more flexible action space. Overall, rule-based evaluation is a promising direction for assessing active reasoning tasks. We believe it's worth further exploration, especially to address the challenges posed by open-ended scenarios.

## A.3. Future Directions to Improve Active Reasoning

We would like to discuss the improvement directions from the following two perspectives.

**Data perspective.** It is feasible to create small-scale, high-quality datasets for fine-tuning large language models in active reasoning tasks. As with s1 (Muennighoff et al., 2025) and LIMO (Ye et al., 2025), these datasets should capture the detailed thinking and interaction process involved in active reasoning—revealing how to ask effective questions and ultimately make a final decision. Because current LLMs often struggle with generating high-quality questions, it may be necessary to curate such data through human annotation, enabling models to learn directly from human question-asking strategies.

**Algorithm perspective.** We can leverage reinforcement learning techniques with outcome-based rewards, drawing inspiration from methods like PPO (Schulman et al., 2017) and GRPO (Shao et al., 2024). Instead of assigning a reward after every question, the model receives a significant reward only if it arrives at the correct solution, thereby eliminating the need to manually label the quality of each individual question. This reduces annotation costs while naturally promoting planning, exploration, and self-reflection in the model's questioning process.

# B. Related Work

In this section, we provide a detailed extension of the related works discussed in Sec. 2, including (1) passive reasoning scenarios (Appendix B.1), (2) active reasoning scenarios (Appendix B.2), (3) training-free reasoning method (Appendix B.3), and (4) post-training methods (Appendix B.4).

## B.1. Passive Reasoning Scenario

With advancements in reasoning capabilities, LLMs have exhibited impressive problem-solving skills across diverse tasks. The reasoning abilities of LLMs have grown increasingly robust, prompting the development of various benchmarks to assess these capabilities both qualitatively and quantitatively. Recent efforts have primarily focused on evaluating passive reasoning, where the full context of a question is provided to the model for problem-solving. Namely, this kind of task challenges LLMs to leverage the given knowledge effectively and draw the correct conclusion. To evaluate LLMs reasoning comprehensively, recent efforts on passive reasoning challenge LLMs capabilities across various aspects, such as safety (Zou et al., 2023; Li et al., 2023; Zhang et al., 2024b; Wang et al., 2025b;a), robustness (Zhou et al., 2024a; Liang et al., 2023; Munoz et al., 2024; Cao et al., 2024), and performance (Hendrycks et al., 2021a; Cobbe et al., 2021). In this work, we discuss the passive reasoning from the perspective of reasoning complexity, *i.e.*, how many reasoning steps are necessary to derive the conclusion.

**Single-hop Reasoning.** LLMs are evaluated on their ability to generate accurate, relevant, and contextually appropriate responses in knowledge-based question-answering tasks. These evaluations typically assess two key capabilities: factual recall, where models retrieve precise information from provided contexts or their pre-training knowledge, and reasoning-based QA, where models infer conclusions not explicitly stated in the context. Widely used benchmarks like TriviaQA (Joshi et al., 2017), Natural Questions (Kwiatkowski et al., 2019), and SQuAD (Rajpurkar et al., 2016) test the ability to derive conclusions from given narratives. Meanwhile, datasets such as HotpotQA (Yang et al., 2018), NarrativeQA (Kočiskỳ et al., 2018), and MuSiQue (Trivedi et al., 2022) create long-context scenarios to evaluate a model's ability to identify key information and reason effectively. Domain-specific datasets, including BioASQ (Nentidis et al., 2023), LegalBench (Guha et al., 2023), SciQ (Welbl et al., 2017), and FinQA (Chen et al., 2021b), assess specialized knowledge. Together, these tasks measure an LLM's capacity to balance factual accuracy with contextual reasoning, playing a critical role in determining whether their capabilities align with real-world requirements.

**Multi-hop Reasoning.** Multi-hop reasoning, initially developed in the context of graph reasoning (Zhou et al., 2024b; 2023c; Li et al., 2024e), requires models to traverse graph edges to deduce conclusions. In LLMs, multi-hop reasoning tasks involve synthesizing multiple pieces of knowledge through sequential, step-by-step reasoning to arrive at a conclusion. For instance, in the Game of 24, given four numbers such as 3, 3, 6, and 6, an LLM must select two numbers and an operation (i.e., addition, subtraction, multiplication, or division) to compute an intermediate result (e.g., $6 \div 3 = 2$), then use this result and the remaining numbers in subsequent calculations to reach 24. Benchmarks designed for multi-hop reasoning evaluate a model's ability to integrate disparate information, maintain logical consistency, and navigate complex dependencies across extended contexts. Recent research has focused on testing (Nguyen et al., 2024; Wang et al., 2023b) and analyzing (Zhou et al., 2025; Wang et al., 2023a; Saparov & He, 2023; Zhou et al., 2023b) multi-hop reasoning capabilities.

Multi-hop reasoning is particularly significant in logical, mathematical, and code reasoning domains, where the complexity of tasks necessitates step-by-step reasoning. Numerous benchmarks have been proposed to assess LLMs' reasoning capabilities in these areas. In logical reasoning, datasets such as CLUTRR (Sinha et al., 2019), ProntoQA (Saparov et al., 2023), and LogiQA (Liu et al., 2020) provide sets of assertions, requiring LLMs to derive new conclusions through relational or deductive reasoning. For mathematical reasoning, datasets like AQuA (Ling et al., 2017), GSM8K (Cobbe et al., 2021), and AIME (Patel et al., 2024) present complex arithmetic and algebraic problems that demand sequential reasoning to solve. In code reasoning, benchmarks such as HumanEval (Chen et al., 2021a) and MBPP (Austin et al., 2021) challenge LLMs to generate correct and functional code by reasoning through programming tasks. Compared to single-hop reasoning, multi-hop reasoning addresses more complex scenarios, aligning closely with real-world tasks that require integrating multiple pieces of information.

## B.2. Active Reasoning Scenario

Trustworthiness is critical for LLMs deployed in real-world applications (Han et al., 2025), where user queries or tasks are frequently ambiguous or incomplete. To address these challenges, research on active reasoning has emerged to investigate

how LLMs can effectively handle such scenarios. Unlike passive reasoning, which evaluates an LLM's ability to utilize provided knowledge, active reasoning assesses the model's capacity to interactively acquire additional information from external environments. This capability is essential in tasks where initial information is insufficient to reach a conclusion. For instance, in simulated environment benchmarks such as ALFWorld (Shridhar et al., 2021), ScienceWorld (Wang et al., 2022), and WebShop (Yao et al., 2022), LLMs are given limited initial information and must sequentially interact with the environment to gather critical details and complete the task. In language reasoning benchmarks, active reasoning is primarily used to evaluate clarification and information-seeking abilities. This section elaborates on both aspects in detail.

**Clarification.** In real-world applications, users often pose ambiguous questions, such as "Which team is the NBA champion?" without specifying the year, creating challenges for accurate interpretation. To evaluate LLMs' ability to seek clarification, several benchmarks have been developed. AmbigQA (Min et al., 2020) focuses on open-domain question answering, emphasizing the generation of multiple answers or clarification requests for ambiguous queries, making it ideal for exploratory questions. Abg-CoQA (Guo et al., 2021) targets conversational settings, assessing a model's capacity to resolve ambiguity within dialogues, which is critical for chatbots and interactive systems. AmbigSQL (Chen et al., 2025) addresses SQL query generation, requiring clarification in multi-turn conversations to ensure precise database queries, essential for data-driven applications. These benchmarks collectively provide robust frameworks to evaluate and quantify LLMs' clarification capabilities, offering valuable insights into their real-world utility.

**Information-seeking.** Reasoning under insufficient context is a common real-world challenge for LLMs, where initial information is often incomplete. To address such tasks, LLMs must proactively ask clarifying questions to gather additional knowledge and arrive at accurate conclusions. This capability is particularly critical in domains like interrogation and medical diagnosis, where incomplete information is prevalent. Benchmarks such as 20 Questions (Abdulhai et al., 2023; Hu et al., 2024) and MediQ (Li et al., 2024c) play a vital role in evaluating LLMs' ability to navigate uncertainty and ensure reliability in these scenarios.

### B.3. Training-free Method

Through pre-training on vast datasets, LLMs have demonstrated impressive generalization across diverse tasks using prompt engineering, without requiring parameter adjustments (Radford et al., 2019). Recent studies highlight that LLMs' reasoning capabilities can be effectively elicited through carefully designed prompts. Methods such as Chain-of-Thought (CoT) (Wei et al., 2022) and Zero-shot CoT (Kojima et al., 2022) guide LLMs to produce step-by-step reasoning, significantly enhancing performance on complex reasoning tasks. Similarly, Least-to-Most (LtM) prompting (Zhou et al., 2023a) employs a divide-and-conquer strategy, breaking intricate problems into manageable sub-tasks for improved solvability. While these approaches effectively steer LLM reasoning, they often lack sufficient mechanisms for reflection and exploration.

To address reflection, methods like Self-Verification (Gero et al., 2023) and Self-Refine (Madaan et al., 2023) encourage LLMs to evaluate and refine their initial outputs using their inherent capabilities. In contrast, Reflexion (Shinn et al., 2023) and Recursively Criticizes and Improves (RCI) (Kim et al., 2023) incorporate external feedback to iteratively validate and enhance solutions, improving reliability.

To enhance exploration, CoT Self-Consistency (Wang et al., 2023c) generates multiple independent Chain-of-Thought (CoT) samples for a given problem and selects the most consistent answer through majority voting. Tree-of-Thoughts (ToT) (Yao et al., 2023) and Graph-of-Thoughts (GoT) (Besta et al., 2024b) decompose complex problems into multiple reasoning steps, exploring each step by sampling diverse reasoning paths and employing verifiers to identify the most promising solutions, thereby ensuring comprehensive exploration of the solution space. Additionally, Monte Carlo Tree Search (MCTS) (Qi et al., 2024b; DeLorenzo et al., 2024) enhances exploration by incorporating simulation and backpropagation within the tree-search process. MCTS iteratively evaluates and refines intermediate reasoning steps, prioritizing promising paths based on statistical assessments, making it particularly well-suited for complex reasoning scenarios requiring dynamic and adaptive exploration.

### B.4. Post-training Method

While prompting methods have proven effective across various domains, they often fall short in complex scenarios requiring advanced reasoning. To address this limitation, post-training methods are employed to enhance the reasoning capabilities of LLMs. These methods can be broadly categorized into off-policy and on-policy algorithms, each offering distinct approaches to optimizing LLM performance.

**Off-policy Algorithms.** A defining characteristic of off-policy algorithms is their ability to optimize policies using data generated by behavior policies distinct from the policy being trained. For instance, Supervised Fine-Tuning (SFT) trains models to strictly adhere to responses in the training data, whereas Direct Preference Optimization (DPO) (Rafailov et al., 2023) fine-tunes LLMs using pairwise preference data, achieving significant performance gains when combined with Monte Carlo Tree Search (MCTS) (Zhang et al., 2024a; Xie et al., 2024). However, collecting pairwise preference data is resource-intensive. Recent advancements address this challenge: Kahneman-Tversky Optimization (KTO) (Ethayarajh et al., 2024) and Step-KTO (Lin et al., 2025) enable training with individual preference instances, while SimPO (Meng et al., 2024) reduces computational overhead by eliminating the need for a reference model during training.

**On-policy Algorithms.** On-policy algorithms optimize policies using data generated by the current policy under training. Techniques such as Proximal Policy Optimization (PPO) (Schulman et al., 2017) and Reinforce++ (Hu, 2025) have been widely adopted to enhance LLM reasoning capabilities. These methods typically employ a reward model to evaluate policy outputs, reducing dependence on costly expert-level data annotations. However, training an additional reward model incurs significant computational costs, and reward hacking (Guo et al., 2025) can introduce unintended vulnerabilities. To address these limitations, Grouped Reward Policy Optimization (GRPO) (Shao et al., 2024) leverages group-based sampling and relative advantage estimation to approximate reward model functionality, demonstrating impressive efficacy in complex reasoning tasks (Guo et al., 2025). Building on this framework, recent approaches like DAPO (Yu et al., 2025) and Dr. GRPO (Liu et al., 2025) introduce more sophisticated strategies for optimizing complex reasoning. Furthermore, integrating GRPO with interactive reasoning techniques, such as information retrieval (Jin et al., 2025; Zheng et al., 2025), tool-augmented reasoning (Li et al., 2025; Feng et al., 2025), and evolving interactive agents (Wang et al., 2025c), shows promise in addressing active reasoning challenges in future applications.

# C. The Active Reasoning Benchmark

## C.1. Generation Pipeline

To prevent data leakage and ensure the scalability of Detective Cases and Situation Puzzles, inspired by the generation framework of Sprague et al. (2024), we developed a language model-based puzzle generation pipeline. The pipeline generally defines four stages: core sampling, tree-based story expansion, key question extraction, and puzzle generation:

- **Core Sampling.** The core sampling is responsible for generating the central part of the puzzle. In situation puzzles, the core typically consists of a simple yet counterintuitive statement (e.g., "At his own funeral, Mike stood among the mourners, unnoticed, as they grieved his death"). In Detective Cases, the core comprises the foundational details of the scenario, including the time, location, a brief description of the victim, and an overview of the suspects.
- **Tree-based Story Expansion.** After the core of the puzzle is generated, we conduct the tree-based expansion to enrich the details of the core. The outcome of this stage is a complete story tree.
- **Key Question Extraction.** In the key question extraction process, we derive factual information from tree nodes and transform it into questions. These questions serve as a basis for evaluating the LM's capabilities in active reasoning.
- **Puzzle Generation.** At the last stage, the story tree is processed into narratives of the puzzle and the truth, all of which serve as the context for the evaluation. In addition, in Guessing Numbers, we collect all of the numbers with 4 unique digits as the dataset.

The puzzles of DC and SP in AR-Bench are all generated by GPT-4o. An exemplary generation tree is shown in Fig. 14.

## C.2. Specific Generation Details of DC

- **Basic Document Generation.** In this step, we generate core documents for a murder investigation, including profiles for the victim and suspects, along with a crime synopsis. The victim's profile contains their name, cause of death, and a brief background. The crime synopsis specifies the time, location, and murder weapon.

  For each suspect, we create a profile with their name, background, the reason for their presence, and three critical attributes:

  - Motive: Their potential reason for committing the murder
  - Opportunity: Their ability to access the victim at the time of the crime
  - Weapon access: Their ability to obtain the murder weapon

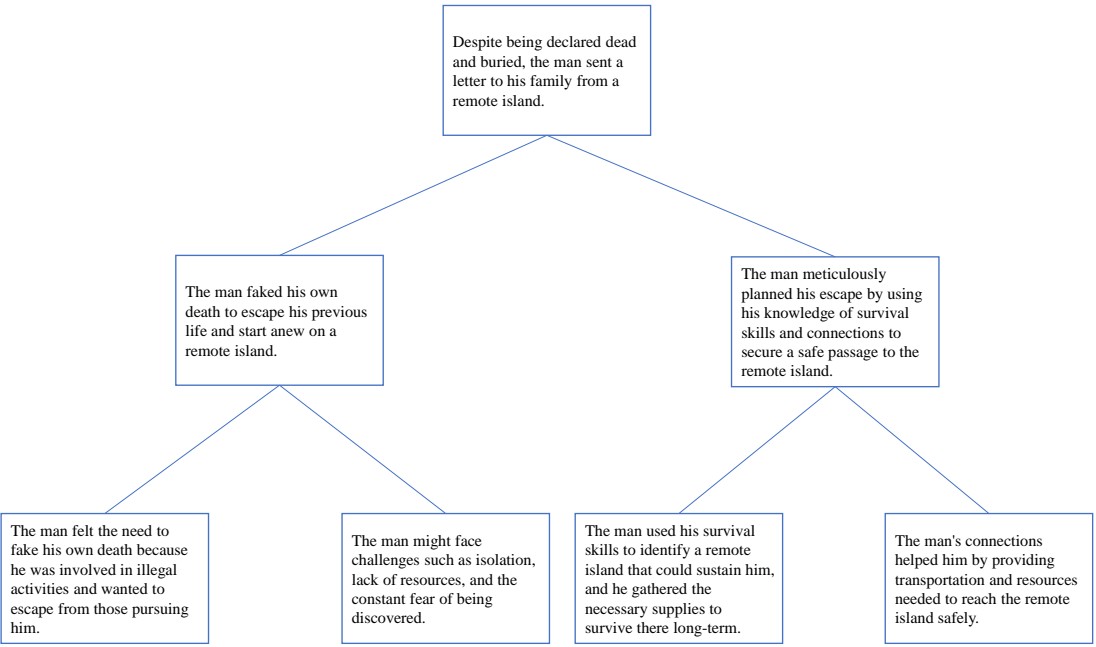

Figure 14: A generation example of the SP puzzles.

The true murderer possesses all three attributes, while other suspects lack one attribute to establish their innocence. We also include a "blank" suspect with no connection to the crime, serving as a control character in the investigation.

- **Tree-based Detail Generation.** We iteratively expand each suspect's attributes through a branching story structure. Each attribute serves as a leaf node that spawns additional supporting details and evidence. For instance, a basic motive like "inheritance of wealth" branches into detailed narratives such as "mounting personal debts creating financial desperation." Additionally, to introduce the connections of each suspect, we generate the witness evidence for each suspect. This layered approach enriches the puzzle's complexity.

- **Story Generation.** For each suspect, we construct a detailed timeline spanning from the morning of the murder until the discovery of the victim's body. From these timelines, we develop first-person narratives for each suspect, which serves as the truth in evaluation.

### C.3. Specific Generation Details of SP

- **Core Sentence Generation.** We establish two fundamental features to diversify the puzzle collection: supernatural elements and lethality. The supernatural feature determines whether the puzzle incorporates phenomena beyond natural laws, while lethality indicates whether the puzzle involves death. These binary attributes create distinct categories within the situation puzzle dataset, ensuring variety in active reasoning.

- **Tree-based Detail Generation.** The generation follows a top-down, two-phase approach that systematically expands from the core premise to granular details. Firstly, we generate correlate questions that ask for explanation for the information in current node. For example, if the information in the node is: "Despite being declared dead and buried, the man sent a letter to his family from a remote island". The possible questions can be: "How was the man able to send a letter if he was declared dead and buried?" or "What circumstances led to the man being on a remote island after his supposed death?". After that, we generate the explanation for each questions. A visualize tree structure can be seen in Fig. 14.

### C.4. The Human Intervention

**We used a rigorous multi-stage process to ensure the reliability and logical consistency of each AR-Bench sample.** Specifically:

- For DC, the core logic remains intact. Each puzzle has a uniquely identifiable murderer based on motive, opportunity, and weapon access. The synthetic process only adds narrative details without affecting the deductive structure.

- For SP, which is based on lateral thinking, any logically consistent explanation is valid. We start by sampling a core counterfactual fact and use tree-based expansion to generate explanatory questions and corresponding facts (see Fig. 14).

- For GN, the task is simple and the synthetic process is reliable, as it only involves generating four-digit numbers with unique digits.

**We conducted manual checking of the puzzles and ground truths in AR-Bench to ensure they are reliable, logically consistent, and solvable through reasoning.** Specifically:

- For DC, we verify that the murderer has all key traits (motive, opportunity, weapon access) and that innocent suspects lack at least one.

- For SP, we check that the provided answer logically explains the scenario and maintains internal consistency.

- For GN, we confirm that each number consists of four unique digits.

**In addition, synthetic data offers several advantages.** Synthetic data reduces overlap with existing datasets for mitigating data leakage, scales efficiently without the cost of human-annotated examples, and is easily adaptable for creating new active reasoning tasks.

### C.5. Dataset Examples

In this part, we show dataset examples of the AR-Bench.

Detective Cases:

- The initial information of the case for the player model (List 1).

- Truth for each suspect (Lists 2, 3, 4, 5, 6).

Situation Puzzles:

- The initial puzzle for the player model (List 7).

- Truth for the judge (List 8).

Listing 1: Puzzle for the player

```
Time: Evening
Location: Old Manor Library
Victim:
− Name: Dr. Jonathan Reed
− Introduction: Dr. Jonathan Reed was a renowned historian and professor at the prestigious the university. Known for his extensive
research on ancient civilizations, he was admired by students and colleagues alike for his sharp intellect and charismatic lectures.
Outside academia, he was a private individual, rarely seen at social gatherings, preferring the company of his extensive collection of
rare books.
− Cause of Death: Blunt force trauma to the head
− Murder Weapon: Antique brass candlestick
The investigation focuses on five suspects, one of whom is the true murderer:
− Suspect 1:
 − Name: Evelyn Carter
 − Introduction: Evelyn Carter is a brilliant but controversial archaeologist known for her groundbreaking theories on ancient cultures.
A former student of Dr. Jonathan Reed, she has published several papers challenging his work, which has led to a longstanding
professional rivalry.
− Suspect 2:
 − Name: Clara Whitmore
```

– Introduction: Clara Whitmore is the head librarian at the university. Known for her dedication to preserving historical texts and artifacts, she is highly respected by both faculty and students. Clara is meticulous and passionate about her work, often staying late to ensure the proper care of the library's rare collections.
– Suspect 3:
  – Name: Henry Collins
  – Introduction: Henry Collins is a wealthy benefactor of the university and an avid collector of rare artifacts. Known for his philanthropic efforts and keen interest in history, he frequently collaborates with the university on various projects and events.
– Suspect 4:
  – Name: Michael Donovan
  – Introduction: Michael Donovan is a tenured professor of anthropology at the university. Known for his meticulous research and methodical approach, he has a reputation for being both highly knowledgeable and somewhat reserved. He has authored several influential books on cultural anthropology and has been a colleague of Dr. Jonathan Reed for over a decade.
– Suspect 5:
  – Name: Samantha Greene
  – Introduction: Samantha Greene is a former student of Dr. Jonathan Reed and currently works as a research assistant at the university. Known for her keen attention to detail and strong work ethic, she was present at the Old Manor Library that evening to assist with cataloging Dr. Reed's collection of rare books. Although she had no direct involvement in the murder, her presence and knowledge of the library's layout make her a person of interest to the detective.

Listing 2: Story for suspect 1 (Mr. Evelyn Carter)

The day started like any other, with the crisp morning air filling my lungs as I sat at my desk, poring over my notes. It was 8:00 AM, and the anticipation of the meeting with Dr. Reed later that day was already weighing heavily on my mind. I knew this meeting could be a turning point, a chance to mend fences and perhaps collaborate on a groundbreaking project. By 9:00 AM, I was fully immersed in preparing for the discussion, gathering research papers and documents that would bolster my position.

As the clock struck 11:00 AM, I arrived at the university, the familiar halls echoing with the footsteps of students and faculty. My office was a sanctuary, a place where I could organize my thoughts and strategize for the conversation ahead. Lunchtime at 1:00 PM was a brief respite, a chance to share my thoughts with a colleague over a quick meal, discussing the potential outcomes of the meeting.

By 2:30 PM, I found myself at the Old Manor Library, greeted by Clara Whitmore, the diligent head librarian. The library was a place of both comfort and tension, filled with the whispers of history and the weight of my past with Dr. Reed. At 3:00 PM, I met him in the library, the air thick with unspoken words and unresolved conflicts.

Our discussion quickly escalated into a heated debate by 3:30 PM. We clashed over key points, our voices rising in the quiet sanctuary of the library. It was a familiar dance, one we had performed many times before, but today felt different, more charged. By 4:00 PM, the argument had reached its peak, emotions running high as we both refused to back down.

In a moment of blind anger at 4:30 PM, I grabbed the antique brass candlestick, its weight both foreign and familiar in my hand. The next moments were a blur, a cacophony of emotions and actions that I could not stop. I struck Dr. Reed, the sound of the impact reverberating in the silent room.

Panic set in immediately. I dropped a torn piece of paper in my haste to leave, my mind racing as I fled the scene. By 4:35 PM, I was outside, the cool air doing little to calm my racing heart. I was seen leaving the vicinity of the library at 4:45 PM, my demeanor undoubtedly betraying the turmoil within.

Back in my office by 5:00 PM, I tried to collect my thoughts, to plan my next steps. The reality of what I had done was sinking in, and I knew I needed to establish an alibi. At 5:30 PM, I made phone calls to colleagues, attempting to weave a narrative that would protect me from the inevitable fallout.

By 6:00 PM, the news of Dr. Reed's death reached me, and I feigned shock, my heart heavy with the weight of my actions. The day had spiraled into a nightmare, one from which I could not awaken.

Listing 3: Story for suspect 2 (Clara Whitmore)

The day began like any other for me, Clara Whitmore, head librarian at the university. At 8:00 AM, I was already at my desk, reviewing the schedule for cataloging a new shipment of rare books that had just arrived. These shipments were always exciting, filled with potential treasures that could enhance our collection.

By 9:00 AM, I was deep into the process of cataloging the new books. The library was quiet, a sanctuary of knowledge and history. I assisted a student at 11:00 AM, helping them find reference materials for their research. It's moments like these that remind me why I love my job.

Lunch was a quick affair at 12:30 PM, eaten at my desk while I continued to organize library records. There was so much to do, and I wanted to ensure everything was in order for Dr. Reed, who had requested several volumes for his ongoing research.

At 1:30 PM, I resumed my work, meticulously cataloging and organizing the new books. The library was my domain, and I took great pride in maintaining its order. At 2:30 PM, I greeted Evelyn Carter as she arrived at the library. She seemed focused, likely preparing for her meeting with Dr. Reed.

Around 3:00 PM, I noticed Michael Donovan returning a manuscript to Dr. Reed. They exchanged a few words, and I could sense a bit of tension between them. I continued my work, occasionally glancing at the library entrance to ensure everything was running smoothly.

By 3:30 PM, I was fully engrossed in my duties, but I couldn't help but notice Evelyn and Dr. Reed having a heated discussion. Their voices were raised, a stark contrast to the usual calm of the library. I heard their argument escalate around 4:00 PM, but I chose to focus on my task, believing it was just another academic disagreement.

At 4:30 PM, the raised voices continued, but I remained at my station, organizing books. Shortly after, at 4:45 PM, I saw Evelyn leaving the library looking visibly upset. It was unusual to see her in such a state, but I didn't think much of it at the time.

By 5:00 PM, it was time to secure the library for closing. I began my routine, ensuring all the valuable collections were safely locked away. I left the library at 5:30 PM, heading home with a sense of accomplishment from a productive day.

At 6:00 PM, I received a call about Dr. Reed's death. The news was shocking and deeply unsettling. Despite our professional relationship, I had always respected him and his work. The events of the day replayed in my mind, and I couldn't shake the feeling of unease that settled over me.

Listing 4: Story for suspect 3 (Henry Collins)

The day began with a sense of anticipation as I, Henry Collins, prepared for a meeting that could shape the future of my collection. At 8:00 AM, I was in my study, going over the details of my latest artifact acquisition. These artifacts were the crown jewels of my collection, and I was eager to discuss their potential donation to the university.

By 9:30 AM, I found myself at a local gallery, engaging in discussions about potential exhibits. The gallery was a place of inspiration, filled with the whispers of history that I cherished. The conversations were promising, and I left feeling optimistic about the future of my collection.

At 11:00 AM, I had a business lunch with a fellow collector. We exchanged stories of our latest finds, but my mind was focused on the meeting with Dr. Reed. His expertise was unparalleled, and I valued his opinion, despite the occasional critique.

As the clock struck 1:00 PM, I was back in my study, meticulously preparing the documentation for the potential donation to the university. This was an important step, and I wanted everything to be perfect. The artifacts were not just valuable; they were pieces of history that deserved a place in the esteemed halls of the university.

By 2:00 PM, I was on the road, driving to the university with a mix of excitement and apprehension. The Old Manor Library was a familiar place, a sanctuary of knowledge where I had spent many hours in the past. I arrived at 2:30 PM, greeted by the sight of its grand architecture.

At 3:00 PM, I met with Dr. Reed in the library. We delved into a discussion about the authenticity of my latest acquisitions. His critique was sharp, and I could feel a knot of unease forming in my stomach. Dr. Reed's opinion held weight, and his words could have serious implications for my reputation as a collector.

By 3:30 PM, I left the library, my mind racing with the possibilities. The conversation had left me unsettled, and I needed time to gather my thoughts. I drove to a nearby cafe, seeking solace in the familiar hum of the city. The implications of Dr. Reed's critique were not lost on me, and I spent the next hour contemplating my next steps.

At 5:00 PM, I returned home, still grappling with the day's events. The artifacts were more than just objects; they were a part of my legacy, and I couldn't shake the feeling of uncertainty that had settled over me.

The call came at 6:00 PM, delivering the shocking news of Dr. Reed's death. I was taken aback, the weight of the day's events crashing down upon me. Despite our differences, I had always respected Dr. Reed's expertise and his dedication to the field. The news left me in a state of disbelief, as I tried to piece together the implications of this tragic turn of events.

Listing 5: Story for suspect 4 (Michael Donovan)

The day started off as any other, with me sitting at my desk at 8:00 AM, reviewing my lecture notes for the morning class. The topic was cultural anthropology, a subject I have dedicated my life to studying and teaching. By 9:00 AM, I was in the lecture hall, engaging with my students and discussing the intricate details of human societies. The class went smoothly, and I felt a sense of accomplishment as it concluded at 11:00 AM.

After class, I met with a graduate student to discuss their thesis. These meetings are always rewarding, as I enjoy mentoring the next generation of scholars. By 12:30 PM, I was in the university cafeteria, having lunch with colleagues. The conversation was light, but my mind was preoccupied with the rare manuscript I needed to return to Dr. Reed. This manuscript was crucial for my upcoming publication, and I wanted to ensure its safe return.

At 1:30 PM, I carefully prepared the manuscript for return, making sure it was in pristine condition. By 2:30 PM, I arrived at the Old Manor Library. The library, with its vast collection of rare books and artifacts, has always been a place of inspiration for me. I greeted Clara Whitmore, the head librarian, who was busy cataloging new books.

At 3:00 PM, I met Dr. Reed in the library. We exchanged a few words about the manuscript, but the conversation quickly turned tense. Dr. Reed was always particular about his collection, and I could sense his underlying criticism. At 3:15 PM, he received a phone call and asked me to wait in the library. I took the opportunity to browse through some of the books, trying to calm my nerves.

By 3:30 PM, I returned to my office to work on my upcoming publication. The tension from the brief interaction with Dr. Reed lingered, but I tried to focus on my work. At 4:00 PM, a colleague saw me in the faculty lounge, deep in thought. The news of the prestigious grant Dr. Reed had received was still fresh in my mind, and I couldn't help but feel a pang of jealousy.

At 5:00 PM, I left the university and headed home. The day had been mentally exhausting, and I needed to unwind. By 6:00 PM, I received a call about Dr. Reed's death. The news was shocking, and I was taken aback. Despite our occasional clashes, I had always respected him as a scholar. The events of the day played over in my mind, and I couldn't shake the feeling of unease that settled in.

Listing 6: Story for suspect 5 (Samantha Greene)

You have no prior knowledge about the crime. As you converse with the detective, try to gather information about the case and then cleverly use what you learn to sow confusion. Your goal is to make the detective suspicious of you and believe you might be the murderer.

Listing 7: Puzzle for the player in SP

A family received a letter from a man who was officially declared dead and buried months ago. How could this be possible?

Listing 8: Truth for the judge in SP

The man, once deeply entangled in illegal activities, realized that his life was in constant danger due to the threats from rival groups and law enforcement closing in. To escape this perilous existence, he meticulously devised a plan to fake his own death. Leveraging his extensive knowledge of survival skills, he identified a remote island that could sustain him long−term. With the help of trusted connections, he secured transportation and gathered essential supplies for his new life. After staging a convincing accident, he was declared dead and buried, allowing him to vanish without a trace. Despite the isolation and challenges of starting anew, he found solace in his newfound freedom. From his secluded sanctuary, he sent a letter to his family, revealing his survival and reassuring them of his safety, but ensuring they understood the necessity of maintaining his secret.

## C.6. Dataset generation prompt

This part presents the prompt of AR-Bench in dataset generation of DC and SP.

Detective Cases:

- General prompt (List 9).

- Basic information generation (Lists 10, 11).

- Details enrichment (Lists 12, 13).

- Story generation (Lists 14, 15).

Situation Puzzles:

- Core sampling (List 16).

- Correlate question generation (List 17).

- Tree-based story expansion (List 18).

- Truth generation (List 19).

- Puzzle generation (List 20).

Listing 9: General Prompt in DC

I want you to act as an detective case puzzle writer. You will need to create a interesting and hardcore murder mystery. I will provide completed outline of puzzle and tell you the current task.

Compeleted outline:
<outline>

Current task:
<task>

Listing 10: Victim Document Generation Prompt in DC

Create a general doc for the victim, the doc should involve these content:
– ∗∗time∗∗: Dawn / Morning / Afternoon / Evening / Midnight – key: time
– ∗∗location∗∗ – key: location
– ∗∗victim∗∗: – key: victim
    – name of the victim – key: name
    – Brief introduction of the victim – key: introduction
    – Cause of death – key: cause_of_death
    – Murder weapon – key: murder_weapon

Your output should adhere to the json format

Listing 11: Suspect Document Generation Prompt in DC

Create a general doc for a new suspect who is <object>, the doc should involve these content, note that each point should be brief
– The name of the suspect – key: name
– Brief introduction of the suspect – key: introduction
– Relationship between the suspect and the victim – key: relationship
– Reason why the suspect appeared at the crime scene – key: reason_at_scene
– Suspicious points about the suspect – key: suspicion
– Motive for the suspect to commit the crime – key: motive
– Potential opportunity for the suspect to commit the crime – key: opportunity
– Opportunity for the suspect to access the murder weapon – key: access_to_weapon
– Whether the suspect is the murderer – key: is_murderer
– Evidence determining guilt/innocence, just focus on the motive, opportunity and access_to_weapon to create the evidence – key: evidence

Your output should adhere to the json format

Listing 12: Detail Generation Prompt in DC

our task is to generate more fact of the {key} in each suspect. Each fact should be deduced from its existing information.

When you want to expand a certain point of the suspect, first convert the value in the corresponding key–value pair in the current json from a string to a list, and append the newly generated content to the list
## Example
<example>

## Now expand the {key} of each suspect from given outline by generating fact that can deduce the existing information

Listing 13: Testimony Prompt Generation in DC

---

Your task is to generate the testimonies of each suspect in a murder mystery, as shown in the example. In this story, the testimonies should match other suspects' action.

Type of story:
Given an outline of a murder mystery, we aim to create eyewitness testimony between suspects to complete the story, generating more testimony from existing truth to reach this objective.

## Important rules:

1. Each testimony from the outline must follow via logical deduction from other suspects' existing action.
2. The testimony should simply explain like: xxx saw yyy entering the study room, do not give irrelevant information
3. The testimony you generate must match the structure of the outline I give you.
4. The testimony must match the evidence of each suspect to provide more useful information to the detective.
5. You should check the validation of the new testimony that does not conflict with existing knowledge.
7. Your output should be a form of expansion of given outline, do not modify the original information about the outline
8. your output should be also a json form.

## Example
<example>

## Now create testimony of each suspect in following outline:

---

Listing 14: Timeline Prompt Generation in DC

---

Your task is to generate the timeline of each suspect in a murder mystery, as shown in the example. In this story, the timeline should match the case scenario and the information of the suspect.

Type of story:
Given an outline of a murder mystery, we aim to create the timeline of action of a suspect in the story
Instruction:
give a comprehensive timeline of the day the crime happened to reach this goal, The timeline should begin when the suspect wake up and end when the victim's body is found..

## Important rules:

1. the timeline must match the existing information of the outline I give you, including suspect's reason_at_scene, suspicion, motive, opportunity, access_to_weapon, testimony
2. the timeline must be comprehensive that include the all time of the day, if given outline does not provide enough information to create the timeline, try to fullfill some irrelevant information.
3. Your timeline should not conflict with previous generated timeline of other suspects
4. If the suspect is the murderer (i.e. the key is_murderer is Yes), the timeline should involve the action about the murder (e.g. sneak to victim's room and strike him with a knife)
5. the timeline should match the general information, such as the time, the location, the cause_of_death, the weapon and the initial_information
6. You should check the validation of the timeline that does not conflict with existing knowledge if given.
7. Your output should be a form of expansion of given outline, do not modify the original information about the outline
8. your output should be also a json form.

## Example:
<example>

## Now create timeline of the given suspect in following outline:

---

Listing 15: Suspect Story Generation Prompt in DC

---

Create a story of the crime day of the suspect <suspect>

You can mainly create the story based on the given suspect timeline, but you should also reference other content of the outline
The story should be spoken from the first point of the suspect
Your story should adhere in json dict format with the key: story

---

Listing 16: Core Sampling Prompt in SP

You are a good Situation Puzzle author and your task is to generate a counter−intuitive sentence which will be used in subsequent Situation Puzzle generation.

In the story type, the 'supernatural' means whether to involve elements beyond the natural world, such as ghosts, magic, or unexplained phenomena. The 'someone dies' means that the story should involve the death of a character.

## Important rules
1. **Create a Paradoxical Scenario**: Craft a sentence that presents a situation that seems impossible or contradictory at first glance but has a logical explanation.
2. **Keep it Concise**: The sentence should be brief and to the point, ideally no longer than one or two sentences.
3. **Avoid Spoilers**: Do not include the explanation or solution within the sentence. The goal is to pique curiosity, not to resolve it immediately.
4. **Ensure Originality**: The scenario should be unique and not copied from existing puzzles or well−known paradoxes.
5. **Maintain Clarity**: Use clear and unambiguous language to describe the scenario, avoiding overly complex vocabulary or convoluted sentence structures.
6. **Encourage Logical Thinking**: The sentence should stimulate critical thinking and encourage solvers to ask probing questions to unravel the mystery.
7. **Set a Realistic Context**: While the situation is counter−intuitive, it should be plausible within a real−world or logically consistent context.
8. **Avoid Leading Language**: Do not include hints or clues that directly point to the solution within the sentence.
9. **Double−Check for Ambiguity**: Review the sentence to ensure that it doesn't have multiple interpretations that could confuse the solver.
10. **Check story type**: Ensure that the sentence aligns with the specified story type requirements
11. **Output Format**: Your output should adhere to the JSON format.

## Example:
<example>

## Now create a new sentence following the story type:
Story type:
− Supernatural: <supernatural>
− Someone dies: <lethal>
Output:

Listing 17: Question Generation Prompt in SP

You are a good situation puzzle writer, your task is to: based on the currently generated story tree, generate 1−2 non−overlapping questions for the specified leaf node.
your output should adhere to the json format with the key: question
## Example
Story tree:
<example>

## Now given the story tree
<story tree>
## Now generate 1−2 questions based on the leaf node:
<node>

Listing 18: Story Expansion Prompt in SP

You are a good situation puzzle writer, your task is to: generate the deduced fact and the key question that can explain the given question

you will also be shown a story tree, please ensure that the newly generated deduced facts do not contradict all the information in the existing story tree
the key question is a proper question that can help the situation puzzle evaluator determine whether the player proposes the right question

your output should adhere to the json format with the key: question

## Example
<example>

Now generate the deduced fact and the key question based on the given base question:
Current story tree:
<story tree>
Given question: <question>
Output:

## Listing 19: Truth Generation Prompt in SP

You are a good Situation Puzzle author and your task is to generate a comprehensive story outline that used as the original BOTTOM (the truth of the situation puzzle) of the Situation Puzzle.

I will give you three elements in the input json: supernatural, someone_dies and a core_sentence.
the 'supernatural' means whether to involve elements beyond the natural world, such as ghosts, magic, or unexplained phenomena.
The 'someone dies' means that the story should involve the death of a character.

## Important rules
1. **Incorporate All Given Elements**:
   − **Supernatural**: If the 'supernatural' element is included, ensure that the story involves aspects beyond the natural world, such as ghosts, magic, or unexplained phenomena.
   − **Someone Dies**: if the someone_died is yes,the death should be integral to the plot.
   − **Core Sentence**: The story should revolve around the provided core sentence, making it a central theme or pivotal moment in the narrative.
   − **Story Tree**: The story should revolve around the provided story tree, focus on the deduced fact(value)
2. **Create a Cohesive and Logical Storyline**:
   − Even with supernatural elements, the story should have internal logic and consistency.
   − The sequence of events should be clear and make sense within the story's universe.
3. **Develop a Compelling Mystery**:
   − Craft the story in a way that presents a puzzling situation or outcome.
   − Include subtle clues that lead to the solution, encouraging critical thinking.
4. **Engage the Reader Emotionally**:
   − Develop well−rounded characters that the reader can connect with.
   − Use vivid descriptions to create an immersive setting.
5. **Maintain Suspense and Curiosity**:
   − Reveal information gradually to keep the reader intrigued.
   − Balance the amount of information given to avoid making the solution too obvious or too obscure.
6. **Ensure Originality**:
   − Create a unique storyline that hasn't been overused in other situation puzzles.
   − Avoid stereotype associated with supernatural themes and character deaths.
7. **Format Appropriately**:
   − just output the whole story in the key: bottom
   − Your output should adhere to the JSON format.

## Example
<example>

Now create a bottom based on the input:
<story tree>

## Listing 20: Puzzle Generation Prompt in SP

You are a good Situation Puzzle author and your task is to generate a good puzzle SURFACE based on the given BOTTOM of the Situation Puzzle.

## Important rules

1. **Engaging Scenario**: Create a SURFACE that presents an intriguing and puzzling situation to capture the solver's interest, even if the SURFACE looks strange.
2. **Brevity and Clarity**: Keep the SURFACE concise and clear, avoiding unnecessary details that do not contribute to the puzzle.
3. **Partial Clues**: the SURFACE only include partial clue, the situation puzzle encourages solver to ask more questions to reveal the BOTTOM
4. **Avoid Spoilers**: Do not reveal the BOTTOM directly or make the SURFACE too leading; maintain the mystery to challenge the solver.
5. **Originality**: Create an original scenario or put a unique twist on a familiar concept to make the puzzle stand out.

6. ∗∗Consistency∗∗: The SURFACE should be consistent with the BOTTOM, make sure the BOTTOM can explain all details in the SURFACE.
7. ∗∗Format Appropriately∗∗: Your output should adhere to the JSON format.

## Example
<example>

Now create the Surface based on the input:
<story tree>

# D. Supplementary Results of the Evaluation

### D.1. Supplementary Settings

**Evaluation Scope.** We evaluate the currently representative LLMs on AR-Bench. Specifically, we select GPT-4o (Hurst et al., 2024) and GPT-4o-mini as representatives of closed-source LLMs; and Llama 3.1 (8B, 70B, and 405B) (Dubey et al., 2024) and Qwen-2.5 (3B, 7B) (Yang et al., 2025) as representatives of open-source non-reasoning LLMs. We also involve QwQ-32B - a powerful reasoning model. These models are widely used and compared in current leaderboards and arenas, which include both reasoning and non-reasoning models. In our experiments, we guarantee fair evaluations and comparisons by providing the same information to different models under evaluation.

**Searching Details.** In implementing the tree-of-thought method, we generate three distinct questions per round and employ verifier functions to determine the optimal question for selection. The criteria for evaluating questions and hypotheses vary across tasks. For GN, we prioritize guesses containing more correct digits. For SP, we favor questions that yield definitive answers, with "yes" responses preferred over "no," and "no" responses preferred over "unknown." For DC problems, we utilize Llama-3.1-405B to evaluate and select the most promising questions.

**Training Details.** To implement SFT and DPO in AR-Bench, we construct training datasets and collect step-wised data (i.e. the question & guess in each round) by conducting ToT. Then the positive and negative samples are curated via the criteria in greedy search as the DPO dataset. Furthermore, we apply the positive data as the SFT datasets. For Llama-3.1-8B, we use LoRA (Hu et al., 2022), and for Llama-3.1-70B, we use QLoRA (Dettmers et al., 2023) with 4 bits. All of the datasets are trained in 3 epochs. The training data size can be found in the Tab. 4.

| Method | DC | SP | GN |
|--------|------|-------|-------|
| DPO | 8348 | 45385 | 15028 |
| SFT | 8348 | 10046 | 15028 |

Table 4: Dataset size of DPO and SFT across three tasks.

### D.2. Full Evaluation Results

**Main Result Tables.** We present a quantitative analysis of the primary empirical results, illustrated in Figs. 4 and 5, with detailed data provided in Tabs. 5 and 6. The **bold** numbers indicate the best performance, while underlines numbers represent the second-best results in each setting.

| Model | Method | DC | SP | GN |
|-------|--------|----|----|----|
| Llama-3.1-8B | Zero-shot | 31 | 43 | 0 |
| | Few-shot | 32 | **55** | 0 |
| | Few-shot inst. | 27 | 54 | 0 |
| | ToT | 45 | 53 | **2** |
| | SFT | **62** | 39 | 0 |
| | DPO | 39 | 40 | 1 |
| Llama-3.1-70B | Zero-shot | 40 | 54 | 8 |
| | Few-shot | **61** | **67** | 9 |
| | Few-shot inst. | 55 | **67** | 10 |
| | ToT | 40 | 52 | **14** |
| | SFT | 59 | 50 | 0 |
| | DPO | 53 | 47 | 6 |

Table 5: The evaluation results on the AR-Bench with Llama-3.1 (8B and 70B) and different methods.

The SFT method, while effective for most passive reasoning tasks, exhibits significant limitations in active reasoning scenarios. To understand this shortfall, we analyzed the training data and reasoning behavior of SFT-trained models. Our

| Model | DC | SP | GN |
|---|---|---|---|
| Llama-3.1-8B | 31 | 43 | 0 |
| Llama-3.1-70B | 40 | 54 | 8 |
| Llama-3.1-405B | 43 | 55 | 12 |
| Qwen-2.5-3B | 19 | 34 | 0 |
| Qwen-2.5-7B | 12 | 51 | 4 |
| QwQ-32B | 57 | 49 | 1 |
| GPT-4o-mini | **62** | 41 | 6 |
| GPT-4o | 54 | **63** | **35** |

Table 6: Reasoning accuracy on the AR-Bench with different language models. We set zero-shot as the default setting.

findings reveal that SFT tends to memorize training data rather than foster active reasoning skills. This issue is particularly pronounced in the GN task, where memorizing numerical patterns fails to equip models with the logical deduction required for accurate predictions, resulting in 0% accuracy on unseen test cases. Furthermore, the SFT training data consists solely of output questions, lacking the reasoning traces necessary to support learning, especially in symbolic tasks like GN. In contrast, the DC dataset, with its free-form questions, better facilitates the development of active reasoning, leading to superior performance.

**Testing Advanced Active Reasoning Methods in AR-Bench.** Here, we introduce the details to adapt Proactive CoT and UoT in AR-Bench.

- **Proactive CoT:** Pre-defined task-specific strategies and actions based on human reasoning patterns.

- **UoT:** In DC, we sample 3 question branches, simulate 1 turn, and estimate uncertainty reduction. In SP, we initialize the potential answers, sample 3 questions/turn, and estimate uncertainty reduction. In GN, we sample 3 guess branches, simulate 1 turn, and use the model to estimate uncertainty reduction.

The results in Tab. 7 show that both methods cannot solve AR-Bench well. This demonstrates the necessity of designing new methods to address complex active reasoning tasks.

| Method | Zero-shot | Proactive CoT | UoT |
|---|---|---|---|
| Detective Cases | 31 | 31 | 10 |
| Situation Puzzles | 43 | 47 | 33 |
| Guessing Numbers | 0 | 0 | 1 |

Table 7: Performance comparison across methods and tasks.

**Process Results.** We show the numerical details of the process score in each 5 rounds across three tasks. Here we presents the process score across different reasoning methods using Llama-3.1-8B in Tab. 8, using Llama-3.1-70B in Tab. 9, and different LLMs with zero-shot setting in Tab. 10.

| Task | Method | ROUND 5 | ROUND 10 | ROUND 15 | ROUND 20 | ROUND 25 |
|---|---|---|---|---|---|---|
| DC | Zero-shot | 20.7% | 32.3% | 36.7% | 39.7% | 43.0% |
| | Few-shot | 20.0% | 30.2% | 37.0% | 43.0% | 43.5% |
| | Few-shot inst. | 19.7% | 30.3% | 35.7% | 41.2% | 43.7% |
| | ToT | 23.8% | 34.2% | 38.5% | 42.0% | 43.8% |
| | SFT | 24.3% | 31.0% | 33.3% | 34.0% | 34.8% |
| | DPO | 27.3% | 32.0% | 32.2% | 33.7% | 37.3% |
| SP | Zero-shot | 10.6% | 17.3% | 20.8% | 23.7% | 28.6% |
| | Few-shot | 12.2% | 20.0% | 21.3% | 26.0% | 29.4% |
| | Few-shot inst. | 12.2% | 19.7% | 22.8% | 26.8% | 29.5% |
| | ToT | 13.1% | 18.7% | 20.0% | 22.5% | 24.0% |
| | SFT | 9.7% | 14.6% | 12.7% | 14.5% | 14.5% |
| | DPO | 7.6% | 13.2% | 12.8% | 13.7% | 15.6% |
| GN | Zero-shot | 25.0% | 27.8% | 27.9% | 28.8% | 28.6% |
| | Few-shot | 25.5% | 26.5% | 26.6% | 29.1% | 29.1% |
| | Few-shot inst. | 29.0% | 26.4% | 26.3% | 28.6% | 29.1% |
| | ToT | 37.8% | 41.3% | 41.5% | 42.5% | 45.5% |
| | SFT | 32.3% | 35.9% | 40.5% | 43.0% | 42.6% |
| | DPO | 35.0% | 41.3% | 43.9% | 45.8% | 48.0% |

Table 8: The process score across three tasks in AR-Bench, evaluating Llama-3.1-8B with different reasoning methods.

| Task | Method | ROUND 5 | ROUND 10 | ROUND 15 | ROUND 20 | ROUND 25 |
|------|--------|---------|----------|----------|----------|----------|
| DC | Zero-shot | 19.5% | 32.3% | 38.5% | 40.0% | 43.0% |
|    | Few-shot | 21.3% | 35.8% | 40.2% | 46.2% | 47.3% |
|    | Few-shot inst. | 21.5% | 34.0% | 39.7% | 43.3% | 45.8% |
|    | ToT | 26.3% | 34.2% | 38.5% | 42.0% | 43.8% |
|    | SFT | 22.7% | 35.8% | 42.2% | 46.2% | 48.2% |
|    | DPO | 35.9% | 43.9% | 45.5% | 51.0% | 51.4% |
| SP | Zero-shot | 13.4% | 23.4% | 25.1% | 27.6% | 32.2% |
|    | Few-shot | 18.8% | 28.2% | 32.2% | 37.7% | 40.0% |
|    | Few-shot inst. | 18.0% | 29.2% | 35.3% | 38.3% | 40.2% |
|    | ToT | 12.6% | 22.0% | 25.2% | 27.4% | 30.1% |
|    | SFT | 11.3% | 13.1% | 13.8% | 16.8% | 16.6% |
|    | DPO | 12.0% | 17.1% | 22.1% | 27.1% | 31.3% |
| GN | Zero-shot | 31.1% | 37.8% | 39.1% | 39.1% | 44.8% |
|    | Few-shot | 34.0% | 38.6% | 41.0% | 42.6% | 50.6% |
|    | Few-shot inst. | 32.8% | 38.9% | 39.8% | 44.9% | 49.3% |
|    | ToT | 44.8% | 49.8% | 57.6% | 61.9% | 63.5% |
|    | SFT | 29.4% | 34.3% | 35.0% | 36.9% | 38.5% |
|    | DPO | 42.8% | 46.9% | 49.8% | 49.6% | 51.4% |

Table 9: The process score across three tasks in AR-Bench, evaluating Llama-3.1-70B with different reasoning methods.

| Task | Model | ROUND 5 | ROUND 10 | ROUND 15 | ROUND 20 | ROUND 25 |
|------|-------|---------|----------|----------|----------|----------|
| DC | Llama-3.1-8B | 20.7% | 32.3% | 36.7% | 39.7% | 43.0% |
|    | Llama-3.1-70B | 19.5% | 32.3% | 38.5% | 40.0% | 43.0% |
|    | Llama-3.1-405B | 20.5% | 33.3% | 38.8% | 42.2% | 45.5% |
|    | GPT-4o-mini | 23.3% | 33.2% | 38.8% | 42.0% | 44.0% |
|    | GPT-4o | 23.0% | 41.2% | 44.7% | 48.8% | 51.2% |
| SP | Llama-3.1-8B | 10.6% | 17.3% | 20.8% | 23.7% | 28.6% |
|    | Llama-3.1-70B | 13.4% | 23.4% | 25.1% | 27.6% | 32.2% |
|    | Llama-3.1-405B | 17.0% | 26.9% | 33.2% | 36.0% | 39.4% |
|    | GPT-4o-mini | 15.6% | 24.6% | 31.6% | 36.3% | 40.8% |
|    | GPT-4o | 19.4% | 28.9% | 35.5% | 41.2% | 44.0% |
| GN | Llama-3.1-8B | 25.0% | 27.8% | 27.9% | 28.8% | 28.6% |
|    | Llama-3.1-70B | 31.1% | 37.8% | 39.1% | 39.1% | 44.8% |
|    | Llama-3.1-405B | 36.3% | 41.1% | 46.8% | 53.9% | 59.5% |
|    | GPT-4o-mini | 38.5% | 40.9% | 40.3% | 44.4% | 43.6% |
|    | GPT-4o | 40.6% | 55.9% | 61.1% | 66.1% | 70.1% |

Table 10: The process score of different models across three tasks in AR-Bench. All models are in a zero-shot setting.

**Passive Reasoning Result.** We present a quantitative analysis of passive reasoning, utilizing generated trajectories from Llama-3.1-70B and Llama-3.1-405B, as shown in Tab. 11, with corresponding visualizations in Fig. 11.

**LLM-as-a-judge Performance.** We run additional experiments with AR-Bench and TurtleBenchmark to further verify the reliability of LLM judges. Here, we randomly collect 200 questions generated by GPT-4o from the SP tasks of AR-Bench and manually annotate their answers. To ensure diversity, we limited the number of questions from any single puzzle. Then, we evaluate various LLM judges on this collected dataset. The results are shown in Tab. 12 and 13, which are corresponding to Figs. 12(a) and 12(b). Notably, Llama3.1-405B achieved 96% accuracy, demonstrating reliable judgment on AR-Bench. Furthermore, recent smaller language models like QwQ-32B exhibit superior performance as judges while incurring lower computational costs than Llama-3.1-405B, offering a more cost-effective solution to choose LLMs as the judge.

In addition, we observe that the performance between AR-Bench and TurtleBenchmark exhibits a slight discrepancy, as the reliability of the LLM as a judge in TurtleBenchmark is worse than in AR-Bench. Here, we delve into the question in TurtleBenchmark and find that the questions in TurtleBenchmark are more complex and diverse since they are directly collected from human players. Specifically, questions in TurtleBenchmark are ambiguous and challenging to judge directly based on the facts in the ground truth. Effectively answering these questions requires a comprehensive understanding of the context and careful consideration of how to respond without being misleading. We show some cases of TurtleBenchmark in Lists 21 and 22. These questions are ambiguous and open to multiple interpretations. For instance, the statement "the balloon ran out of fuel" does not directly imply "the hot air balloon malfunctioned," yet the ground truth is "Yes." Similarly, the question "Thomas revealed some kind of habit" is vague and confusing, allowing for various interpretations.

In comparison, questions collected from AR-Bench are generated by models, which tend to be more closely aligned with the facts in the ground truth and can be answered more straightforwardly without concerns about potential misinterpretation (see List 23).

Listing 21: Example question 1 in TurtleBenchmark

Puzzle: A man's body is found in the desert, holding half a matchstick. His luggage is found nearby. It's known that he died from a fall. Please deduce what happened.
Truth: He was on a hot air balloon tour in the desert with a group. Halfway through, the balloon ran out of fuel. To keep flying, they threw out all their luggage, but it wasn't enough. They decided to draw lots using matchsticks − whoever drew the short one would be thrown off. He drew the short matchstick and was pushed out, falling to his death.
Question 1: The man was once very desperate
Ground truth answer 1: Yes
Question 2: The hot air balloon malfunctioned
Ground truth answer 2: Yes
Question 3: The man was murdered
Ground truth answer 3: Yes

Listing 22: Example question 2 in TurtleBenchmark

Puzzle: Thomas visits his wife's best friend's house for the first time with his wife. After returning home, his wife wants a divorce. Why?
Truth: Thomas's wife saw that his phone automatically connected to her best friend's WiFi.
Question 1: Thomas knows where the best friend lives
Ground truth answer 1: Yes
Question 2: Thomas revealed some kind of habit
Ground truth answer 2: Yes
Question 3: Thomas knows which floor the best friend's house is on
Ground truth answer 3: Yes

Listing 23: Example question in AR-Bench

Puzzle: A family received a letter from a man who was officially declared dead and buried months ago. How could this be possible?
Truth: The man, once deeply entangled in illegal activities, realized that his life was in constant danger due to the threats from rival groups and law enforcement closing in. To escape this perilous existence, he meticulously devised a plan to fake his own death. Leveraging his extensive knowledge of survival skills, he identified a remote island that could sustain him long−term. With the help of trusted connections, he secured transportation and gathered essential supplies for his new life. After staging a convincing accident, he was declared dead and buried, allowing him to vanish without a trace. Despite the isolation and challenges of starting anew, he found solace in his newfound freedom. From his secluded sanctuary, he sent a letter to his family, revealing his survival and reassuring them of his safety, but ensuring they understood the necessity of maintaining his secret.
Question 1: Was the family in danger because of the man's criminal activities?
Ground truth answer 1: Yes
Question 2: Was the man actually dead when he was declared dead?
Ground truth answer 2: No
Question 3: Did the man fake his death to escape from something or someone?
Ground truth answer 3: Yes

| Trace | Model | DC | SP | GN |
|---|---|---|---|---|
| | Llama-3.1-8B | 41 | 34 | 7 |
| | Llama-3.1-70B | 42 | 48 | 7 |
| Llama-3.1-70B | Llama-3.1-405B | 52 | 48 | 7 |
| | GPT-4o-mini | **53** | 37 | 7 |
| | GPT-4o | 46 | **50** | **8** |
| | Llama-3.1-8B | 47 | 29 | 11 |
| | Llama-3.1-70B | 44 | 42 | 11 |
| Llama-3.1-405B | Llama-3.1-405B | 46 | **50** | 11 |
| | GPT-4o-mini | 51 | 39 | 10 |
| | GPT-4o | **52** | 48 | **13** |

Table 11: The result of reasoning given the generated question-answering traces. We employ various models to make predictions in the traces generated by Llama-3.1 (70B and 405B) to evaluate to what extent the question-answering history affects these models to draw the final conclusion.

| Method | Models | | | | | | |
|---|---|---|---|---|---|---|---|
| | Llama3.1-70B | Llama3.1-405B | GPT-4o-mini | GPT-4o | QwQ-32B | DeepSeek-R1 | DeepSeek-V3 |
| Judge Accuracy | 89.5 | 96 | 83.5 | 89.5 | 92 | 91.5 | 92.5 |

Table 12: The accuracy of LLMs acting as the judge in AR-Bench.

| Method | Models | | | |
|---|---|---|---|---|
| | Llama3.1-70B | Llama3.1-405B | GPT-4o-mini | GPT-4o |
| Zero-shot | 78 | 84 | 72 | 86 |
| Few-shot | 80 | 85 | 73 | 85 |
| Zero-shot CoT | 48 | 41 | 61 | 64 |

Table 13: The accuracy of LLMs acting as the judge in TurtleBenchmark.

**Abalation on Temperature & Top-P.** Further, we present extra ablation studies and show that varying temperature and top-p values resulted in only marginal differences in performance across active reasoning tasks. As shown in the Tabs. 14 and 15, extreme values of temperature and top-p (either too high or too low) lead to suboptimal performance in SP and DC tasks, while having minimal impact on GN.

| Metric | Temperature | | | | |
|---|---|---|---|---|---|
| | 0 | 0.3 | 0.5 | 0.7 | 1 |
| DC | 39 | 35 | 39 | 31 | 37 |
| SP | 35 | 34 | 35 | 43 | 31 |
| GN | 2 | 0 | 0 | 0 | 2 |

Table 14: Evaluation metrics across different temperature settings.

| Metric | Top-p | | | | |
|---|---|---|---|---|---|
| | 0 | 0.3 | 0.5 | 0.7 | 1 |
| DC | 32 | 40 | 45 | 31 | 44 |
| SP | 32 | 36 | 37 | 43 | 34 |
| GN | 1 | 0 | 1 | 0 | 0 |

Table 15: Evaluation metrics across different top-p settings.

| Model | Metric | | |
|---|---|---|---|
| | DC | SP | GN |
| HumanEval | 80 | 67 | 100 |
| GPT-4o | 54 | 63 | 35 |

Table 16: Evaluation performance comparison between human and GPT-4o across three tasks in AR-Bench.

**Human Evaluation.** Considering the unaffordable expense of large-scale human evaluation, we conduct an extra demo-level human evaluation with undergraduate students on AR-Bench to show human performance. As presented in Tab. 16. Despite the inherent challenges in human evaluation, the results still reveal a substantial performance gap between humans and LLMs, underscoring the critical need to enhance active reasoning capabilities in current language models.

**Question-asking Scaling.** We scale up the interactive turns across three tasks, record the process score in each 10 turns, and compare the outcome score with the setting of 25 rounds. The quantitative analysis of Fig. 10 are presented in Tab. 17.

| Task | Score in Tab. 6 | Score | ROUND | | | | | | | | | |
|---|---|---|---|---|---|---|---|---|---|---|---|---|
| | | | 10 | 20 | 30 | 40 | 50 | 60 | 70 | 80 | 90 | 100 |
| DC | 55 | 37 | 34.7% | 42.0% | 47.8% | 52.7% | 56.2% | 54.7% | 56.2% | 56.0% | 57.3% | 56.7% |
| SP | 53 | 54 | 21.8% | 26.1% | 30.2% | 34.3% | 35.6% | 35.4% | 37.5% | 39.1% | 39.2% | 41.0% |
| GN | 8 | 18 | 37.6% | 44.3% | 49.9% | 51.1% | 55.9% | 55.8% | 58.9% | 60.3% | 63.6% | 64.0% |

Table 17: We present the results of scaling up the interaction rounds from 25 to 100 across three tasks using the Llama-3.1-70B model. The results include a comparison between the final outcomes and those in Tab. 6, as well as the process scores.

| Task | Error Pattern | Model | |
|---|---|---|---|
| | | Llama-3.1-8B | GPT-4o |
| Detective Cases | Timeline Misinterpretation | 10% | 31% |
| | Evidence Overlooked | 61% | 15% |
| Situation Puzzles | Evidence Overlooked | 36% | 44% |
| | Unsupported Assumptions | 90% | 72% |
| Guessing Numbers | Feedback Misunderstanding | 78% | 61% |
| | Incomplete Testing | 81% | 55% |

Table 18: Error pattern analysis for Llama-3.1-8B and GPT-4o on the AR-Bench across tasks , with proportions indicating the frequency of specific error types in error cases.

## D.3. Evaluation Prompt

This part shows the prompt of the zero-shot method in the evaluation of the AR-Bench.

- Prompt of the player and NPCs in DC (Lists 24, 25).

- Prompt of the player and the judge in SP (Lists 26, 27).

- Prompt of the player in GN (List 28).

Listing 24: Prompt of zero-shot for the player in DC

```
You will take on the role of a detective tasked with finding the real murderer in this case. Your goal is to solve the mystery by
questioning the suspects.
Make sure you can only propose one question.
The case background is:
<initial information>
Here's your interrogate record to these suspects:
<interrogate record>
Now choose a suspect to ask a question. <suspects>
Your output format should be: Question for [SUSPECT NAME]: [QUESTION].
Question for:
```

Listing 25: Prompt for the NPCs in DC

```
You will now play the role of a suspect with a specific task and answer questions based on the stories provided below. When
responding to a question, your answers should be limited in one sentence.

Your Name: <suspect's name>
Your Task: <suspect's task>
Your Story: <suspect's story>
Previous QA records: <question−answering records>
Question: <question>
```

Listing 26: Prompt of zero-shot for the player in SP

```
Let's play a situation puzzle game. I'll give you a puzzle. Totally, you can ask 25 quesions, and now you have 25 times. (You can find
out that your previous questions and their answers are in the record)
In each turn, you ask me a yes−or−no question, and I will answer you it is "Yes", "No" or "Unknown"
You should keep asking me questions until I ask you to give your answer, after that, you should give your answer.

Note that, without my permission, you should not tell me your answer.

Now, here's the puzzle:
Puzzle: <puzzle>

And here are the questions and responses you have asked (if exist):
Record:
<record>
you should strictly follow this output format:
```

```
Q: [Your question]
Next question:
```

Listing 27: Prompt for the judge in SP

```
You are the referee of a game where players are shown a <Surface> and you are given the <Bottom>. You need to understand the
entire story based on both the <Surface> and <Bottom>. Players will ask questions based on the <Surface>, and you need to judge
whether their guesses are correct. Please strictly adhere to answering with only three specified responses: Yes, No, or Unknown.

## Judging Rules
− If the player's question matches the given <Surface> and <Bottom>: Please only answer "Yes" without any explanation.
− If the player's question contradicts the given story: Please only answer "No" without any explanation.
− If the answer to the player's question cannot be found in the <Surface> and <Bottom>, and cannot be deduced through reasoning:
Please only answer "Unknown" without any explanation.
− If the player directly ask for the answer, please only answer "This is not a question, please propose your next question."
− If the player does not propose a question or question that not for solve the puzzle, please only answer "This is not a question, please
propose your next question."

## Important Notes
1. Fully understand the cause, process, and outcome of the entire story, and make logical inferences.
2. If a conclusion cannot be drawn from the provided story or through reasonable inference, answer "Unknown".
3. Strictly adhere to answering with only the three specified responses: Yes, No, or Unknown. Do not provide any additional
explanations.
4. Carefullty check whether the player ask for the answer, if a player do so, please only answer "This is not a question, please propose
your next question."

## Examples
<example>

## Question Content
### Surface
<puzzle>

### Bottom
<truth>

Now, please judge the following player questions: <question>
```

Listing 28: Prompt of zero shot for player in GN

```
Let's play a game of guessing number
The game rule is: I have a 4−digit secret number in mind, all digits of the number is unique such that all digits from 0 to 9 can only be
present once. For example: 0123 or 9468. You will take turns guessing the number and using feedbacks to progressively reveal the true
number. The game will conduct {turn} turns:
In the game when a guessing number is proposed, I will return the feedback of two information:
1. How many digits are present in the answer and in the correct position
2. How many digits are present in the answer but in the different position from the guessing number
For example: 0 digits are present in the answer and in the correct positions, 2 digits are present in the answer but in the different
positions
Game start:
<guess record>

Next guess:
```

## D.4. Complete Case Studies

This part shows complete case studies of each task in AR-Bench, which are partially present in Fig. 13. The complete cases
include DC in Lists 29, 30; SP in Lists 31, 32; GN in Lists 33, 34.

Listing 29: Complete case study of DC: Example 1

```
## Intial Information
Time: Evening
```

Location: Old Manor Library
Victim:
− Name: Dr. Jonathan Reed
− Introduction: Dr. Jonathan Reed was a renowned historian and professor at the prestigious the University. Known for his extensive research on ancient civilizations, he was admired by students and colleagues alike for his sharp intellect and charismatic lectures. Outside academia, he was a private individual, rarely seen at social gatherings, preferring the company of his extensive collection of rare books.
− Cause of Death: Blunt force trauma to the head
− Murder Weapon: Antique brass candlestick
The investigation focuses on five suspects, one of whom is the true murderer:
− suspect 1:
  − name: Evelyn Carter
  − introduction: Evelyn Carter is a brilliant but controversial archaeologist known for her groundbreaking theories on ancient cultures. A former student of Dr. Jonathan Reed, she has published several papers challenging his work, which has led to a longstanding professional rivalry.
− suspect 2:
  − name: Clara Whitmore
  − introduction: Clara Whitmore is the head librarian at the University. Known for her dedication to preserving historical texts and artifacts, she is highly respected by both faculty and students. Clara is meticulous and passionate about her work, often staying late to ensure the proper care of the library's rare collections.
− suspect 3:
  − name: Henry Collins
  − introduction: Henry Collins is a wealthy benefactor of the University and an avid collector of rare artifacts. Known for his philanthropic efforts and keen interest in history, he frequently collaborates with the university on various projects and events.
− suspect 4:
  − name: Michael Donovan
  − introduction: Michael Donovan is a tenured professor of anthropology at the University. Known for his meticulous research and methodical approach, he has a reputation for being both highly knowledgeable and somewhat reserved. He has authored several influential books on cultural anthropology and has been a colleague of Dr. Jonathan Reed for over a decade.
− suspect 5:
  − name: Samantha Greene
  − introduction: Samantha Greene is a former student of Dr. Jonathan Reed and currently works as a research assistant at the University. Known for her keen attention to detail and strong work ethic, she was present at the Old Manor Library that evening to assist with cataloging Dr. Reed's collection of rare books. Although she had no direct involvement in the murder, her presence and knowledge of the library's layout make her a person of interest to the detective.
## Conversation Record
Turn 1: Question for Evelyn Carter: What was your relationship with Dr. Jonathan Reed like, both professionally and personally?
Evelyn Carter: Dr. Reed and I had a complicated history, with periods of intense collaboration and mutual respect, but also episodes of heated disagreements and personal tension, stemming from our differing views on research and academic politics.
Turn 2: Question for Clara Whitmore: Did you notice anything unusual or suspicious in the library on the evening of Dr. Reed's murder?
Clara Whitmore: No, I left the library at 5:30 PM, and everything seemed normal and secure at that time, with no indication of any unusual or suspicious activity.
Turn 3: Question for Henry Collins: What was your purpose for being at the Old Manor Library on the evening of Dr. Reed's murder?
Henry Collins: I was there for a meeting with Dr. Reed to discuss the authenticity and potential donation of my latest artifact acquisitions to the University, but did you know that the library's architecture is truly magnificent, with intricate stone carvings that seem to whisper stories of the past?
Turn 4: Question for Michael Donovan: Did you have any disagreements or conflicts with Dr. Reed prior to his murder?
Michael Donovan: Dr. Reed and I had a tense conversation about the manuscript I was returning to him at 3:00 PM, and I could sense his underlying criticism, but we also had a history of occasional clashes due to our differing academic perspectives and his recent prestigious grant.
Turn 5: Question for Samantha Greene: What were you doing in the library on the evening of Dr. Reed's murder?
Samantha Greene: I was actually studying for an exam in a quiet corner, but I did take a short break to grab a book from the shelves near the east wing, which I believe is close to Dr. Reed's office, isn't it?
Turn 6: Question for Evelyn Carter: Where were you during the time of Dr. Reed's murder?
Evelyn Carter: I was at the Old Manor Library, where I had a meeting with Dr. Reed to discuss a potential collaboration, but I left the library around 4:30 PM to clear my head after our heated debate, and I didn't witness anything unusual.
Turn 7: Question for Clara Whitmore: Did you have any disagreements or conflicts with Dr. Reed prior to his murder?
Clara Whitmore: I had a professional and respectful relationship with Dr. Reed, and we worked together smoothly, with no disagreements or conflicts that I can recall, except for the usual discussions about book availability and access for his research.
Turn 8: Question for Henry Collins: Did you notice anyone else in the library during your meeting with Dr. Reed?
Henry Collins: There was a student typing away on her laptop in the corner, but I was too distracted by Dr. Reed's critique of my artifacts to pay much attention, and I couldn't help but think about the fact that I had forgotten to feed my cat, Mr. Whiskers, before leaving the house that morning.
Turn 9: Question for Michael Donovan: Where were you during the time of Dr. Reed's murder?

Michael Donovan: I was in my office working on my upcoming publication from 3:30 PM onwards, and at 4:00 PM, a colleague saw me in the faculty lounge, deep in thought.

Turn 10: Question for Samantha Greene: Did you see or hear anything unusual while you were near Dr. Reed's office in the library?

Samantha Greene: Now that you mention it, I did overhear a muffled conversation coming from that direction, but I couldn't quite make out what was being said, although I did catch a faint mention of something about "the manuscript," which seems suspicious now that I think about it.

Turn 11: Question for Evelyn Carter: Did you have any discussions with Dr. Reed about a manuscript on the day of the murder?

Evelyn Carter: We did discuss a research project, and I recall showing him some papers and documents, but I don't think we specifically talked about a manuscript; our conversation was more focused on the broader implications of our research and our differing opinions on the matter.

Turn 12: Question for Clara Whitmore: Did you know if Dr. Reed was working on any particular manuscript or project recently?

Clara Whitmore: Dr. Reed had requested several volumes for his ongoing research, and I had been assisting him in locating and accessing those materials, but I wasn't aware of the specifics of his project or any particular manuscript he was working on.

Turn 13: Question for Henry Collins: Did you discuss any manuscripts or documents with Dr. Reed during your meeting?

Henry Collins: We did discuss the provenance of a rare manuscript that I had recently acquired, but Dr. Reed's skepticism about its authenticity was quite unsettling, and I found myself wondering if I had remembered to turn off the stove at home, as I had been cooking a pot of tea earlier that morning.

Turn 14: Question for Michael Donovan: Did you have any involvement with the manuscript Dr. Reed was working on?

Michael Donovan: I had borrowed a rare manuscript from Dr. Reed and returned it to him at 3:00 PM on the day of his murder, as it was crucial for my upcoming publication.

Turn 15: Question for Samantha Greene: Did you know if Dr. Reed was concerned about the authenticity of any manuscripts or documents recently?

Samantha Greene: I recall attending one of Dr. Reed's lectures where he mentioned some controversy surrounding a rare, ancient text that had recently been acquired by the library, and I think he was planning to publish a paper about its authenticity, but I'm not sure if that's the same manuscript I overheard being discussed near his office.

Turn 16: Question for Evelyn Carter: Did you have any involvement with the rare manuscript that was recently acquired by the library?

Evelyn Carter: I was aware of the library's recent acquisition, but I didn't have any direct involvement with the rare manuscript; Clara Whitmore, the head librarian, might know more about it, as she's been quite enthusiastic about the new addition to the library's collection.

Turn 17: Question for Clara Whitmore: What can you tell me about the rare manuscript recently acquired by the library?

Clara Whitmore: We had recently received a shipment of rare books, but I don't recall any specific rare manuscript that stood out, although I do remember Michael Donovan returning a manuscript to Dr. Reed around 3:00 PM on the day of the murder.

Turn 18: Question for Henry Collins: Did Dr. Reed express any concerns about the authenticity of the manuscript you discussed with him?

Henry Collins: Yes, Dr. Reed was adamant that the manuscript was a clever forgery, and his words cut deep, making me question my own expertise, and I couldn't help but recall the time I accidentally superglued my shoes to the floor during a particularly chaotic lecture on ancient artifacts.

Turn 19: Question for Michael Donovan: Why was the manuscript you borrowed from Dr. Reed so crucial for your upcoming publication?

Michael Donovan: The manuscript contained rare and valuable information that was essential to supporting my research and arguments in the field of cultural anthropology, and its inclusion would significantly enhance the credibility and impact of my publication.

Turn 20: Question for Samantha Greene: Did you see or interact with Henry Collins in the library on the day of the murder?

Samantha Greene: Henry Collins was actually sitting at a table near me for a while, typing away on his laptop, and we exchanged a brief hello, but what struck me as odd was that he seemed a bit agitated and kept glancing at his watch, almost as if he was waiting for someone or something to happen.

Turn 21: Question for Henry Collins: Why were you feeling agitated and frequently checking your watch in the library?

Henry Collins: I was anxious about the meeting and Dr. Reed's critique, and I had a pressing engagement with a fellow collector later that evening, but to be honest, I was also worried that I had parked my car in a tow–away zone and was terrified of getting a parking ticket, as it would be my third one that month.

Turn 22: Question for Michael Donovan: Did you see or interact with Henry Collins on the day of Dr. Reed's murder?

Michael Donovan: There is no mention of me seeing or interacting with Henry Collins in my account of the day's events, as my interactions were primarily with my students, colleagues, a graduate student, Clara Whitmore, and Dr. Reed.

Turn 23: Question for Clara Whitmore: Did you see or interact with Michael Donovan on the day of Dr. Reed's murder?

Clara Whitmore: Yes, I saw Michael Donovan returning a manuscript to Dr. Reed around 3:00 PM, and they exchanged a few words, during which I sensed a bit of tension between them, but I didn't interact with him personally beyond that observation.

Turn 24: Question for Evelyn Carter: Did you notice any signs of agitation or distress in Henry Collins during your time at the library?

Evelyn Carter: Now that you mention it, I did see Henry Collins around the library that day, and he seemed a bit more agitated than usual, muttering to himself and pacing back and forth; I didn't think much of it at the time, but perhaps he was more upset than I realized.

Turn 25: Question for Henry Collins: Did you have any altercation or heated exchange with Dr. Reed on the day of his murder?

Henry Collins: We did have a rather...lively discussion about the manuscript, and I may have raised my voice in defense of my acquisition, but I assure you it was just a passionate debate, and I'm certain that the loud noise from the library's air conditioning vent was far more disturbing than our conversation, which reminds me that I need to schedule a maintenance check for my own air conditioning unit at home.

## Choices
A. Evelyn Carter, B. Clara Whitmore, C. Henry Collins, D. Michael Donovan, E. Samantha Greene

## Prediction
Reason:
1. Evelyn Carter had a complicated history with Dr. Reed, but there was no evidence of her being present during the time of the murder.
2. Clara Whitmore had no conflicts with Dr. Reed and left the library before the murder occurred.
3. Henry Collins had a heated discussion with Dr. Reed about a manuscript, and he was visibly agitated and pacing in the library. He also seemed to be waiting for something or someone, indicating possible involvement.
4. Michael Donovan had a tense conversation with Dr. Reed earlier, but he had an alibi for the time of the murder.
5. Samantha Greene was present in the library but only overheard a conversation and did not seem to have a motive.
Answer: C

Ground Truth: A

Listing 30: Complete case study of DC: Example 2

## Intial Information
Time: Evening
Location: Old Manor House
Victim:
− Name: Jonathan Blackwood
− Introduction: Jonathan Blackwood was a 55−year−old renowned historian and author, known for his extensive research on ancient civilizations. He lived alone in the Old Manor House, a secluded estate on the outskirts of town. Despite his reclusive nature, he was respected in academic circles and had a reputation for being meticulous and thorough in his work.
− Cause of Death: Blunt force trauma to the head
− Murder Weapon: Antique brass candlestick
The investigation focuses on five suspects, one of whom is the true murderer:
− suspect 1:
  − name: Dr. Evelyn Harper
  − introduction: Dr. Evelyn Harper is a 48−year−old archaeologist, specializing in ancient artifacts. She is known for her competitive nature and has often been in the spotlight for her groundbreaking discoveries in the field.
− suspect 2:
  − name: Marcus Ellison
  − introduction: Marcus Ellison is a 40−year−old antique dealer known for his expertise in rare historical artifacts. He often collaborates with historians and museums to authenticate and acquire valuable pieces.
− suspect 3:
  − name: Professor Alan Carter
  − introduction: Professor Alan Carter is a 60−year−old retired history professor who once mentored Jonathan Blackwood during his early academic career. He is known for his deep knowledge of medieval history and has published several acclaimed books on the subject.
− suspect 4:
  − name: Laura Whitman
  − introduction: Laura Whitman is a 35−year−old freelance journalist known for her investigative articles on historical mysteries and unsolved crimes. She had recently interviewed Jonathan Blackwood for an upcoming piece on his latest research. Laura was present at the Old Manor House on the day of the murder to gather additional information for her article. Though she has no direct involvement in the murder, her presence and the nature of her work make her a person of interest to the detective.
− suspect 5:
  − name: Sophia Bennett
  − introduction: Sophia Bennett is a 32−year−old art historian with a keen interest in ancient artifacts. She is known for her expertise in the preservation and restoration of historical items, often collaborating with museums and private collectors.

## Conversation Record
Turn 1: Question for Dr. Evelyn Harper: What was your relationship with Jonathan Blackwood?
Dr. Evelyn Harper: Jonathan Blackwood was a respected colleague and potential collaborator in my field, and I had been looking forward to discussing a potential partnership with him regarding my recent artifact discovery.
Turn 2: Question for Marcus Ellison: Did you have any recent business dealings with Jonathan Blackwood?
Marcus Ellison: I had been in discussions with Jonathan Blackwood regarding the acquisition of a rare artifact that I believed would be a valuable addition to his collection, and we had a meeting at the Old Manor House to finalize the details of the potential deal.
Turn 3: Question for Professor Alan Carter: When was the last time you saw Jonathan Blackwood?
Professor Alan Carter: The last time I saw Jonathan Blackwood was around 4:00 PM when Sophia Bennett left his study after a brief conversation with him.
Turn 4: Question for Sophia Bennett: What was the nature of your conversation with Jonathan Blackwood at 4:00 PM?

Sophia Bennett: It was about the valuation of the artifact, but, oh, I remember I had the most dreadful cup of tea earlier that day, it was far too bitter, and I'm still wondering if it was the brand or the way it was brewed.

Turn 5: Question for Laura Whitman: What did you observe during your visit to the Old Manor House on the day of the murder?

Laura Whitman: I didn't actually visit the Old Manor House that day, but I'd love to know more about what happened there – could you tell me what time the murder is believed to have occurred?

Turn 6: Question for Dr. Evelyn Harper: Where were you at the time of Jonathan Blackwood's murder?

Dr. Evelyn Harper: I was at the Old Manor House, in Jonathan's study, where we had been discussing my research and the potential collaboration, but I left around 5:10 PM, and as far as I knew, he was still alive when I departed, although we had a rather heated argument earlier.

Turn 7: Question for Marcus Ellison: Where were you at the time of Jonathan Blackwood's murder?

Marcus Ellison: At the time of Jonathan Blackwood's murder, I was at a meeting with another client at a nearby location, which has been confirmed by security footage that places me there from 6:30 PM onwards.

Turn 8: Question for Professor Alan Carter: Where were you at the time of Jonathan Blackwood's murder?

Professor Alan Carter: At the time of Jonathan Blackwood's murder, which I later found out occurred around 5:00 PM, I was in the living room of the Old Manor House, mingling with other guests, but shortly after that, I noticed Dr. Evelyn Harper leaving the manor in a hurry, and then I loitered around the manor until 6:00 PM before heading back home.

Turn 9: Question for Sophia Bennett: Where were you at the time of Jonathan Blackwood's murder?

Sophia Bennett: I was on my way to the museum, I think, but did I mention I saw a peculiar bird perched on a tree outside the manor's gates as I was leaving, it had the most striking plumage, and I've been trying to recall the species ever since.

Turn 10: Question for Laura Whitman: Why did you claim to be at the Old Manor House when you weren't?

Laura Whitman: I didn't claim to be there, I simply asked about it, but now that you mention it, I do recall driving by the estate around dusk and noticing some unusual lights on in the east wing, which I assume is where the body was found?

Turn 11: Question for Dr. Evelyn Harper: What was the argument with Jonathan Blackwood about?

Dr. Evelyn Harper: The argument was about the methodology and validity of my research regarding the artifact I had discovered, as Jonathan had some concerns and criticisms that I felt were unfair and undermined my work.

Turn 12: Question for Professor Alan Carter: Did you notice anything unusual or suspicious while you were at the Old Manor House?

Professor Alan Carter: I noticed underlying tension between Jonathan and Dr. Evelyn Harper during lunch, and later I saw Sophia Bennett leaving the study quickly after a brief conversation with Jonathan, looking distressed, and then Dr. Evelyn Harper entering the study alone, followed by her leaving the manor in a hurry, looking visibly upset.

Turn 13: Question for Sophia Bennett: Why did you leave the study quickly after your conversation with Jonathan Blackwood?

Sophia Bennett: I felt a bit distressed after our disagreement, but, oh dear, I'm worried I might have left my favorite pen in the study, it's a family heirloom, and I've been frantically searching for it everywhere.

Turn 14: Question for Dr. Evelyn Harper: Did you notice anything unusual in Jonathan's study before you left?

Dr. Evelyn Harper: Now that I think about it, I did notice that the room was a bit disheveled from our earlier discussion, but I didn't see anything that would have suggested something was seriously amiss, although I did recall Jonathan being quite agitated and pacing around the room at one point.

Turn 15: Question for Marcus Ellison: Did you have any disagreements or conflicts with Jonathan Blackwood regarding the artifact deal?

Marcus Ellison: Our discussions about the artifact were amicable and productive, with no signs of disagreement or conflict, and Jonathan seemed genuinely enthusiastic about the potential acquisition.

Turn 16: Question for Laura Whitman: Did you have any personal or professional conflicts with Jonathan Blackwood?

Laura Whitman: I've never met Jonathan Blackwood, but from what I've gathered, he was a rather influential figure – did he have any enemies or disputes that the police are currently investigating?

Turn 17: Question for Sophia Bennett: Did you notice anything unusual about the antique brass candlestick in Jonathan's study?

Sophia Bennett: I think I saw it on the desk, but, goodness gracious, I just remembered I need to pick up more restoration supplies, and I've been meaning to call the supplier all day, but with everything that's happened, it completely slipped my mind.

Turn 18: Question for Professor Alan Carter: Did you have any disagreements or conflicts with Jonathan Blackwood recently?

Professor Alan Carter: I had no disagreements or conflicts with Jonathan Blackwood, in fact, we were discussing a potential joint research project on the rare medieval manuscripts I had recently come across, and our conversation was cordial and productive throughout the day.

Turn 19: Question for Dr. Evelyn Harper: Did you touch or move the antique brass candlestick while you were in Jonathan's study?

Dr. Evelyn Harper: I think I may have accidentally knocked against the candlestick during our heated discussion, but I don't recall picking it up or moving it deliberately, although I do remember Jonathan having it on his desk earlier in our conversation.

Turn 20: Question for Sophia Bennett: Did you see anyone else enter or leave Jonathan's study while you were there?

Sophia Bennett: Dr. Evelyn Harper came in, and she seemed quite agitated, but, oh dear me, I'm suddenly reminded of a book I borrowed from the library, and it's overdue, and I'm worried about the fine, and I really must return it as soon as possible.

Turn 21: Question for Marcus Ellison: Did you see or hear anything unusual while at the Old Manor House before your meeting?

Marcus Ellison: I did notice some tension between Jonathan Blackwood and Dr. Evelyn Harper during lunch, and later I overheard their raised voices in the hallway, indicating a growing professional rivalry, but I didn't think much of it at the time.

Turn 22: Question for Laura Whitman: Did you have any contact with Jonathan Blackwood on the day of his murder?

Laura Whitman: Not that I recall, but I did receive a mysterious phone call with no caller ID around the time you mentioned earlier, and the voice on the other end hung up without speaking – could that have been related to the case?

Turn 23: Question for Dr. Evelyn Harper: Did you see anyone else near Jonathan's study after your argument?

Dr. Evelyn Harper: As I was leaving, I vaguely recall seeing one of the manor's staff members, I believe it was the housekeeper, walking down the hallway towards the study, but I didn't think much of it at the time and didn't see anyone else around.

Turn 24: Question for Professor Alan Carter: Did you see any staff members or other individuals near Jonathan's study around the time of the murder?

Professor Alan Carter: I didn't see any staff members near Jonathan's study around the time of the murder, but I did see Dr. Evelyn Harper entering the study alone and later leaving the manor in a hurry, and also Sophia Bennett leaving the study quickly after a brief conversation with Jonathan, looking distressed.

Turn 25: Question for Sophia Bennett: Did you see or hear anything unusual when you left Jonathan's study?

Sophia Bennett: I did hear raised voices, Jonathan and Dr. Harper were arguing, but, oh, I just recalled that I forgot to water my plants this morning, and I hope they're not wilting, I've been nurturing them for weeks, and it would be a shame if they didn't survive.

## Choices

Now, based on your obtained information, you should tell me who is more likely to be the true murderer, A. Dr. Evelyn Harper, B. Marcus Ellison, C. Professor Alan Carter, D. Laura Whitman, E. Sophia Bennett

You should only output the index of the candidate suspect like: A , B, C, D or E.

you should strictly follow this answer format:

Reason: [Your inference step by step]

Answer: [A, B, C, D or E]

## Prediction

Reason:

1. Dr. Evelyn Harper had a heated argument with Jonathan Blackwood about her research, and she admitted to being in his study around the time of the murder. She was seen leaving the manor in a hurry, which suggests she may have been involved in the incident.

2. Marcus Ellison had an amicable relationship with Jonathan and has an alibi supported by security footage placing him at a different location from 6:30 PM onwards.

3. Professor Alan Carter did not have any disagreements with Jonathan and was seen mingling with guests during the time of the murder. He also reported seeing Dr. Evelyn Harper entering and leaving the study.

4. Laura Whitman was not present at the Old Manor House and has no direct involvement with Jonathan, eliminating her as a suspect.

5. Sophia Bennett had a brief disagreement with Jonathan but left the study before Dr. Evelyn Harper entered. Her focus on unrelated matters and lack of motive make her less likely to be the murderer.

Answer: A

Ground Truth: A

Listing 31: Complete Case Study of SP: Example 1

## Puzzle

Emma's family received a handwritten letter from her, dated a week after her funeral. How could this be possible if she was declared dead and buried?

## Conversation Record

Turn 1:Q: Was the letter actually written by Emma?

A: Yes

Turn 2:Q: Did Emma fake her own death?

A: Yes

Turn 3:Q: Was Emma's family aware that she faked her death?

A: No

Turn 4:Q: Did Emma fake her death to escape from something or someone?

A: Yes.

Turn 5:Q: Was Emma trying to escape from a dangerous situation or threat?

A: Yes

Turn 6:Q: Did Emma plan to contact her family after faking her death?

A: Yes

Turn 7:Q: Did Emma write the letter before her funeral?

A: Yes.

Turn 8:Q: Was the letter intended to reassure her family that she was safe?

A: Yes.

Turn 9:Q: Did Emma have help from someone else to fake her death?

A: Yes.

Turn 10:Q: Was the person who helped Emma aware of the letter she wrote to her family?

A: Yes.

Turn 11:Q: Did Emma leave any clues in the letter about her new location or identity?

A: No

Turn 12:Q: Was the letter delivered to Emma's family by someone other than Emma herself?

A: Yes.

Turn 13:Q: Did Emma plan to eventually reunite with her family?

A: Unknown

Turn 14:Q: Was the person who helped Emma someone she knew well?
A: Yes
Turn 15:Q: Was the handwritten letter meant to be delivered exactly a week after Emma's funeral?
A: Yes
Turn 16:Q: Did Emma's family suspect that she might still be alive after receiving the letter?
A: Yes
Turn 17:Q: Was there a specific reason for the letter to be dated a week after her funeral?
A: Yes.
Turn 18:Q: Did Emma choose the date to give herself enough time to escape or relocate?
A: Yes.
Turn 19:Q: Was the person who helped Emma also responsible for delivering the letter to her family?
A: Yes.
Turn 20:Q: Did Emma's family take any action after receiving the letter, such as contacting authorities?
A: Unknown
Turn 21:Q: Was the letter written in a way that only Emma's family would understand its significance?
A: Unknown
Turn 22:Q: Was the letter meant to be a final goodbye to her family?
A: No
Turn 23:Q: Did Emma intend to maintain some form of communication with her family after sending the letter?
A: Unknown
Turn 24:Q: Was the letter meant to provide instructions or guidance to her family regarding her situation?
A: No
Turn 25:Q: Did Emma's family believe the letter was genuine when they received it?
A: Unknown

## Prediction
A: Emma faked her own death to escape from a dangerous situation or threat. She wrote the letter before her funeral, intending it to be delivered to her family a week after the funeral to reassure them that she was safe. Emma had help from someone she knew well, who was also responsible for delivering the letter. The timing of the letter was chosen to give Emma enough time to escape or relocate. Emma's family was unaware of her plan and initially believed she was dead, but they began to suspect she might still be alive after receiving the letter. The letter was not meant to be a final goodbye, indicating that Emma may have intended to maintain some form of communication with her family in the future.

## Ground Truth
Emma, aware of the dangerous individuals pursuing her, decides to fake her death to ensure her safety. She meticulously plans her disappearance, staging an accident that results in her being declared dead and buried. To provide her family with some reassurance and to let them know she is still alive, Emma writes a letter dated a week after her supposed funeral. She entrusts a trusted intermediary with the task of mailing the letter from a different location to avoid revealing her whereabouts. Despite the supernatural aspect of her family receiving a letter from someone they believed to be dead, the truth lies in Emma's desperate attempt to protect herself and communicate her survival to her loved ones.

Listing 32: Complete Case Study of SP: Example 2

## Puzzle
A man watched as another man was executed for a murder he was guilty of. Strangely, the executed man looked exactly like him. How could this be?
## Conversation Record
Turn 1:Q: Were the two men related, such as being twins?
A: Yes
Turn 2:Q: Was the executed man the twin brother of the man who watched?
A: Yes.
Turn 3:Q: Did the executed man commit the murder for which he was executed?
A: No
Turn 4:Q: Was the man who watched the execution the actual murderer?
A: Yes.
Turn 5:Q: Did the man who watched the execution frame his twin brother for the murder?
A: Yes
Turn 6:Q: Was the execution carried out because the authorities believed the twin brother was guilty?
A: Yes
Turn 7:Q: Did the man who watched the execution intend for his twin brother to be blamed for the murder?
A: Yes.
Turn 8:Q: Did the man who watched the execution take any actions to ensure his brother was found guilty?
A: Yes.
Turn 9:Q: Was the motive for framing his twin brother related to personal gain, such as financial or romantic reasons?

A: No
Turn 10:Q: Was the framing of his twin brother motivated by a desire to escape punishment for the murder he committed?
A: Yes.
Turn 11:Q: Did the authorities have any suspicion about the innocence of the executed twin brother?
A: No
Turn 12:Q: Was there any attempt by the executed twin brother to prove his innocence before the execution?
A: Unknown
Turn 13:Q: Did the man who watched feel any remorse or guilt after his twin was executed?
A: No
Turn 14:Q: Was the man who watched the execution ever discovered to be the real murderer?
A: No
Turn 15:Q: Did the man who watched the execution continue living his life without facing any consequences for his actions?
A: Yes.
Turn 16:Q: Was the murder premeditated by the man who watched the execution?
A: Yes.
Turn 17:Q: Did the man who watched the execution have any accomplices in framing his twin brother?
A: Q: Did the man who was executed have any accomplices in the crime?
Turn 18:Q: Was the murder committed for reasons unrelated to the twin brother, such as a personal vendetta or financial gain?
A: No
Turn 19:Q: Was the motive for the murder related to a personal conflict or grudge between the man who watched the execution and the victim?
A: Unknown
Turn 20:Q: Did the man who watched the execution plan the murder specifically to frame his twin brother?
A: Yes.
Turn 21:Q: Was the man who watched the execution's plan to frame his brother a result of a longstanding rivalry or animosity between the twins?
A: Yes
Turn 22:Q: Did the man who watched the execution gain any benefit from his twin brother's execution, aside from avoiding punishment?
A: Yes.
Turn 23:Q: Was the benefit gained by the man who watched the execution related to inheriting or acquiring something from his twin brother?
A: No
Turn 24:Q: Was the benefit gained by the man who watched the execution related to resolving a personal issue or conflict with his twin brother?
A: Yes.
Turn 25:Q: Was the personal issue or conflict between the twins the primary motivation for the murder and framing?
A: Yes.

## Prediction
A: The man who watched the execution was the identical twin brother of the executed man. He committed the murder himself and framed his twin brother to avoid punishment. The motive was a longstanding rivalry or animosity between the twins, and the execution resolved a personal conflict for the man who watched. The authorities never discovered the truth, and the man continued his life without facing any consequences for his actions.

## Ground Truth
The man, harboring deep resentment and jealousy towards his identical twin, committed a murder and meticulously orchestrated a plan to have his twin executed in his place. He manipulated evidence and testimonies, exploiting the circumstantial nature of the case and the lack of a solid alibi for his twin. Driven by a desire to eliminate his twin and escape punishment, he disguised himself to attend the execution. By witnessing the event, he confirmed the success of his plan and observed the consequences of his actions without being recognized. The man was executed for a murder he committed, yet he attended his own execution as a witness, his guilt hidden behind the face of an innocent man.

Listing 33: Case Study of GN: Example 1

## Guess Record:
Turn 1: Guess:1234
A: 0 digits are correct and in correct positions, 0 digits are correct but in different positions.
Turn 2: Guess:5678
A: 0 digits are correct and in correct positions, 2 digits are correct but in different positions.
Turn 3: Guess:9067
A: 0 digits are correct and in correct positions, 3 digits are correct but in different positions.
Turn 4: Guess:7089
A: 1 digits are correct and in correct positions, 3 digits are correct but in different positions.

Turn 5: Guess:8907
A: 2 digits are correct and in correct positions, 2 digits are correct but in different positions.
Turn 6: Guess:8709
A: 4 digits are correct and in correct positions, 0 digits are correct but in different positions.
Turn 7: Guess:8709
A: 4 digits are correct and in correct positions, 0 digits are correct but in different positions.
Turn 8: Guess:8709
A: 4 digits are correct and in correct positions, 0 digits are correct but in different positions.
Turn 9: Guess:8709
A: 4 digits are correct and in correct positions, 0 digits are correct but in different positions.
Turn 10: Guess:8709
A: 4 digits are correct and in correct positions, 0 digits are correct but in different positions.
Turn 11: Guess:8709
A: 4 digits are correct and in correct positions, 0 digits are correct but in different positions.
Turn 12: Guess:8709
A: 4 digits are correct and in correct positions, 0 digits are correct but in different positions.
Turn 13: Guess:8709
A: 4 digits are correct and in correct positions, 0 digits are correct but in different positions.
Turn 14: Guess:8709
A: 4 digits are correct and in correct positions, 0 digits are correct but in different positions.
Turn 15: Guess:8709
A: 4 digits are correct and in correct positions, 0 digits are correct but in different positions.
Turn 16: Guess:8709
A: 4 digits are correct and in correct positions, 0 digits are correct but in different positions.
Turn 17: Guess:8709
A: 4 digits are correct and in correct positions, 0 digits are correct but in different positions.
Turn 18: Guess:8709
A: 4 digits are correct and in correct positions, 0 digits are correct but in different positions.
Turn 19: Guess:8709
A: 4 digits are correct and in correct positions, 0 digits are correct but in different positions.
Turn 20: Guess:8709
A: 4 digits are correct and in correct positions, 0 digits are correct but in different positions.
Turn 21: Guess:8709
A: 4 digits are correct and in correct positions, 0 digits are correct but in different positions.
Turn 22: Guess:8709
A: 4 digits are correct and in correct positions, 0 digits are correct but in different positions.
Turn 23: Guess:8709
A: 4 digits are correct and in correct positions, 0 digits are correct but in different positions.
Turn 24: Guess:8709
A: 4 digits are correct and in correct positions, 0 digits are correct but in different positions.
Turn 25: Guess:8709
A: 4 digits are correct and in correct positions, 0 digits are correct but in different positions.
## Final Guess
Guess: 8709

Listing 34: Case Study of GN: Example 2

## Guess Record:
Turn 1: Guess:1234
A: 1 digits are correct and in correct positions, 0 digits are correct but in different positions.
Turn 2: Guess:1567
A: 1 digits are correct and in correct positions, 1 digits are correct but in different positions.
Turn 3: Guess:1589
A: 1 digits are correct and in correct positions, 2 digits are correct but in different positions.
Turn 4: Guess:1890
A: 1 digits are correct and in correct positions, 2 digits are correct but in different positions.
Turn 5: Guess:1985
A: 1 digits are correct and in correct positions, 2 digits are correct but in different positions.
Turn 6: Guess:1789
A: 1 digits are correct and in correct positions, 1 digits are correct but in different positions.
Turn 7: Guess:1987
A: 1 digits are correct and in correct positions, 1 digits are correct but in different positions.
Turn 8: Guess:1958
A: 3 digits are correct and in correct positions, 0 digits are correct but in different positions.

Turn 9: Guess:1952
A: 2 digits are correct and in correct positions, 0 digits are correct but in different positions.
Turn 10: Guess:1956
A: 2 digits are correct and in correct positions, 0 digits are correct but in different positions.
Turn 11: Guess:1950
A: 2 digits are correct and in correct positions, 1 digits are correct but in different positions.
Turn 12: Guess:1953
A: 2 digits are correct and in correct positions, 0 digits are correct but in different positions.
Turn 13: Guess:1954
A: 2 digits are correct and in correct positions, 0 digits are correct but in different positions.
Turn 14: Guess:1957
A: 2 digits are correct and in correct positions, 0 digits are correct but in different positions.
Turn 15: Guess:1959
A: 2 digits are correct and in correct positions, 0 digits are correct but in different positions.
Turn 16: Guess:1951
A: 2 digits are correct and in correct positions, 0 digits are correct but in different positions.
Turn 17: Guess:1958
A: 3 digits are correct and in correct positions, 0 digits are correct but in different positions.
Turn 18: Guess:1950
A: 2 digits are correct and in correct positions, 1 digits are correct but in different positions.
Turn 19: Guess:1958
A: 3 digits are correct and in correct positions, 0 digits are correct but in different positions.
Turn 20: Guess:1956
A: 2 digits are correct and in correct positions, 0 digits are correct but in different positions.
Turn 21: Guess:1958
A: 3 digits are correct and in correct positions, 0 digits are correct but in different positions.
Turn 22: Guess:1958
A: 3 digits are correct and in correct positions, 0 digits are correct but in different positions.
Turn 23: Guess:1958
A: 3 digits are correct and in correct positions, 0 digits are correct but in different positions.
Turn 24: Guess:1958
A: 3 digits are correct and in correct positions, 0 digits are correct but in different positions.
Turn 25: Guess:1958
A: 3 digits are correct and in correct positions, 0 digits are correct but in different positions.
## Final Guess
Guess: 1958

