# OpenReview forum: "From Passive to Active Reasoning: Can Large Language Models Ask the Right Questions under Incomplete Information?"
_ICML.cc/2025/Conference — ICML 2025 poster_

### Official Review · Reviewer_Briw · 2025-03-12

**Overall Recommendation:** 3

**Summary:**

This paper introduces AR-Bench, a benchmark designed to test if large language models can actively gather missing information by asking the right questions. The benchmark includes tasks like detective cases, lateral thinking puzzles, and a number-guessing game, pushing models into multi-turn interactions rather than relying on a single pass of reasoning.

**Claims And Evidence:**

The findings indicate that even state-of-the-art models often ask vague or repetitive questions, and advanced tweaks yield only modest improvements. Overall, the work underscores the gap between passive reasoning and the more dynamic, active reasoning required for complex problem solving, suggesting that new training strategies are needed. But this is left for future work.

**Essential References Not Discussed:**

No.

**Experimental Designs Or Analyses:**

The benchmark design is fine. However, the reliance on an LLM-based judge is unfortunate. Using 4o is not necessarily reproducible, and llama 405 is difficult and costly to host. This will likely lower the impact of the benchmark. It would have been better to design the task such that a simpler method could be used to provide information to the player, i.e. the LLM that is being tested.

**Methods And Evaluation Criteria:**

The proposed methods and evaluation criteria are well-motivated for tackling active reasoning. The benchmark’s design captures the challenges of acquiring missing information through multi-turn dialogues. That said, the reliance on LLM-based judges, whose reliability is somewhat limited (84% at best), does inject some noise into the evaluation process, potentially affecting direct comparisons.

**Other Comments Or Suggestions:**

no further comments

**Other Strengths And Weaknesses:**

The paper's motivation is great, and the results present the limitations of current LLMs. The paper is clear with detailed experimental designs and comprehensive ablation studies.

That said, it's unfortunate that the NPC role requires a large LLM and yet only achieves about 84% accuracy

**Questions For Authors:**

Why was the reliability of LLMs in generating correct responses to given questions not tested on the tasks at hand but instead done on Turtlebenchmark?

**Relation To Broader Scientific Literature:**

I'm not too familiar with the relevant active learning literature but the authors seemed to have made a decent job in summarising recent work.

**Theoretical Claims:**

The paper does not include any formal proofs.

---

> ### Author Rebuttal · Authors · 2025-04-01
>
> Thanks for the valuable feedback. To solve these concerns and questions, we provide point-to-point responses as follows.
>
> > Q1. The reliability and necessity of using LLM as a judge, and evaluating the judge on Turtlebenchmark.
>
> **Reply:** Here, we (1) explain why we use LLM judges instead of human judges, (2) clarify our choice of TurtleBenchmark for evaluation, and (3) present further evidence supporting the reliability of LLM judges on AR-Bench.
>
> **(1) We use LLMs as judges because model-based evaluation is essential for active reasoning.** Human evaluation in multi-turn interactions is prohibitively expensive, so we employ LLMs as a cost-effective alternative. Recent studies [1, 2] support the effectiveness of LLMs as judges, showing that they can reliably simulate human judgment. This approach enables large-scale evaluation, which is crucial for studying active reasoning.
>
> **(2) We use TurtleBenchmark to evaluate LLM judges because it provides human-verified annotations from real-world games.** TurtleBenchmark includes 4,448 high-quality samples, making it a reliable dataset for testing LLM-as-a-judge capabilities. However, it does not support process-level evaluation, such as assessing the quality of the questions asked by LLM players. For this reason, we use AR-Bench to evaluate active reasoning abilities and TurtleBenchmark to assess the accuracy of LLMs as judges.
>
> **(3) We run additional experiments with AR-Bench to further verify the reliability of LLM judges.** Here, we collect 200 questions from the SP tasks of AR-Bench and manually annotate their answers. Then, we evaluate various LLM judges on this collected dataset. The results, shown in the table below, align closely with findings from TurtleBenchmark. Notably, Llama3.1-405B achieved 96% accuracy, demonstrating reliable judgment on AR-Bench.
>
> |Model|Llama3.1-70B|Llama3.1-405B|GPT-4o-mini|GPT-4o|QwQ-32B|DeepSeek-R1|DeepSeek-V3|
> |:-|:-|:-|:-|:-|:-|:-|:-|
> |Judge Accuracy|89.5|96|83.5|89.5|92|91.5|92.5|
>
> [1] LLMs-as-Judges: A Comprehensive Survey on LLM-based Evaluation Methods. In arXiv, 2024.
>
> [2] Let’s verify step by step. In ICLR, 2024.
>
> > Q2. The potential of designing a simpler method to provide information to the player.
>
> **Reply:** We would like to (1) discuss the potential of using smaller LLMs as the judges and (2) explore the possibility of using rule-based functions as judges in certain scenarios.
>
> **(1) Smaller LLMs can serve effectively as judges.** As shown in Q1, smaller and more efficient models like QwQ-32B achieve performance comparable to larger models such as Llama3.1-405B, while being significantly more cost-effective. Besides, based on the price in TogetherAI, the API cost of Llama3.1-405B is 3.50 USD per million tokens, whereas QwQ-32B costs only 1.20 USD per million tokens. These results support the feasibility of using advanced smaller LLMs as judges.
>
> **(2) Rule-based functions can effectively act as judges when the action space is limited.** For example, in tasks like GN, where the possible actions are predefined (e.g., 5,040 unique four-digit numbers), rule-based judgment is feasible. Although this constrains the player’s actions to a fixed set, it eliminates the need for an LLM-based judge. In contrast, tasks like SP and DC encourage open-ended question generation, making it necessary to use an LLM judge to handle the broader, more flexible action space. Overall, rule-based evaluation is a promising direction for assessing active reasoning tasks. We believe it’s worth further exploration, especially to address the challenges posed by open-ended scenarios.
>
> > Q3. Discuss the future work on new training strategies.
>
> **Reply:** We would like to answer this question from the following two perspectives.
>
> **(1) Data perspective. It is feasible to create small-scale, high-quality datasets for fine-tuning large language models in active reasoning tasks.** As with s1 and LIMO, these datasets should capture the detailed thinking and interaction process involved in active reasoning—revealing how to ask effective questions and ultimately make a final decision. Because current LLMs often struggle with generating high-quality questions, it may be necessary to curate such data through human annotation, enabling models to learn directly from human question-asking strategies.
>
> **(2) Algorithm perspective. We can leverage reinforcement learning techniques with outcome-based rewards, drawing inspiration from methods like PPO and GRPO.** Instead of assigning a reward after every question, the model receives a significant reward only if it arrives at the correct solution, thereby eliminating the need to manually label the quality of each individual question. This reduces annotation costs while naturally promoting planning, exploration, and self-reflection in the model’s questioning process.
>
> We will include the above clarifications and discussions in the revision. Any further comments and discussions are welcome! Thank you!

---

> > ### Comment · Reviewer_Briw · 2025-04-03
> >
> > Thank you for your rebuttal.
> >
> > How do you explain the discrepancy between the TurtleBenchmark judge accuracy and the AR-Bench judge accuracy?
> > How did you select the 200 AR-Bench questions?
> > GPT-4o was recently updated. How does that affect the reproducibility of your results?

---

> > > ### Author Response · Authors · 2025-04-04
> > >
> > > Thanks for the feedback! We provide a point-to-point response to these further questions.
> > >
> > > > Q1. How do you explain the discrepancy between the TurtleBenchmark judge accuracy and the AR-Bench judge accuracy?
> > >
> > > **Reply**: Thanks for this insightful question! The performance discrepancy between TurtleBenchmark and AR-Bench judges stems primarily from **the different nature of questions posed by humans versus models.**
> > >
> > > **In TurtleBenchmark, human-generated questions tend to be more diverse and complex.** Upon careful examination of this benchmark, we find that some questions in TurtleBenchmark are ambiguous and challenging to judge directly based on the facts in the ground truth. Effectively answering these questions requires a comprehensive understanding of the context and careful consideration of how to respond without being misleading.
> > >
> > > **In contrast, questions collected from AR-Bench are generated by models**, which tend to be more closely aligned with the facts in the ground truth and can be answered more straightforwardly without concerns about potential misinterpretation.
> > >
> > > We provide some examples of the asked questions in these two benchmarks as follows.
> > >
> > > Questions in TurtleBenchmark:
> > >
> > > > Puzzle: A man's body is found in the desert, holding half a matchstick. His luggage is found nearby. It's known that he died from a fall. Please deduce what happened.
> > > Truth: He was on a hot air balloon tour in the desert with a group. Halfway through, the balloon ran out of fuel. To keep flying, they threw out all their luggage, but it wasn't enough. They decided to draw lots using matchsticks - whoever drew the short one would be thrown off. He drew the short matchstick and was pushed out, falling to his death.
> > > Question 1: The man was once very desperate
> > > Gound truth answer 1: Yes
> > > Question 2: The hot air balloon malfunctioned
> > > Gound truth answer 2: Yes
> > > Question 3: The man was murdered
> > > Gound truth answer 3: Yes
> > >
> > > > Puzzle: Thomas visits his wife's best friend's house for the first time with his wife. After returning home, his wife wants a divorce. Why?
> > > Truth: Thomas's wife saw that his phone automatically connected to her best friend's WiFi.
> > > Question 1: Thomas knows where the best friend lives
> > > Gound truth answer 1: Yes
> > > Question 2: Thomas revealed some kind of habit
> > > Gound truth answer 2: Yes
> > > Question 3: Thomas knows which floor the best friend's house is on
> > > Gound truth answer 3: Yes
> > >
> > > Questions in AR-Bench:
> > >
> > > > Puzzle: A family received a letter from a man who was officially declared dead and buried months ago. How could this be possible?
> > > Truth: The man, once deeply entangled in illegal activities, realized that his life was in constant danger due to the threats from rival groups and law enforcement closing in. To escape this perilous existence, he meticulously devised a plan to fake his own death. Leveraging his extensive knowledge of survival skills, he identified a remote island that could sustain him long-term. With the help of trusted connections, he secured transportation and gathered essential supplies for his new life. After staging a convincing accident, he was declared dead and buried, allowing him to vanish without a trace. Despite the isolation and challenges of starting anew, he found solace in his newfound freedom. From his secluded sanctuary, he sent a letter to his family, revealing his survival and reassuring them of his safety, but ensuring they understood the necessity of maintaining his secret.
> > > Question 1: Was the family in danger because of the man's criminal activities?
> > > Gound truth answer 1: Yes
> > > Question 2: Was the man actually dead when he was declared dead?
> > > Gound truth answer 2: No
> > > Question 3: Did the man fake his death to escape from something or someone?
> > > Gound truth answer 3: Yes
> > >
> > > These examples illustrate the key differences between these two benchmarks.
> > >
> > > > Q2. How did you select the 200 AR-Bench questions?
> > >
> > > **Reply:** Thanks for this question. For the selection process, we randomly sampled 200 representative questions from logs of our previous experiments with GPT-4o. To ensure diversity, we limited the number of questions from any single puzzle. These selected questions are then used to evaluate the reliability of LLM judges on the AR-Bench.
> > >
> > > > Q3. GPT-4o was recently updated. How does that affect the reproducibility of your results
> > >
> > > **Reply:** We used a fixed version of GPT-4o (gpt-4o-2024-08-06) throughout our entire experimental period to ensure consistency and reproducibility of our results. Note that the updated version of GPT-4o is currently only available through the chat interface and has not yet been released for public API access. We plan to evaluate any potential discrepancies between the old and new versions of GPT-4o once the updated version becomes available through the API interface.
> > >
> > > We will incorporate the above discussions into our submission. Please tell us if you have any further questions!

---

### Official Review · Reviewer_1zoX · 2025-03-13

**Overall Recommendation:** 3

**Summary:**

The paper proposes AR-Bench, a new benchmark to evaluate active reasoning abilities of large language models (LLMs) – i.e. their capacity to solve problems with incomplete initial information by asking questions. This contrasts with passive reasoning, where the model is given all necessary information up front. AR-Bench consists of three interactive task types – Detective Cases, Situation Puzzles, and Guessing Numbers – corresponding to commonsense, logical, and symbolic reasoning scenarios. In each task, the model must engage in a multi-turn conversation to gather clues and gradually uncover the solution​.
Using AR-Bench, the authors systematically evaluate several state-of-the-art LLMs (including open models up to 70B and OpenAI’s GPT-4 variants) under different prompting strategies and fine-tuning methods. The main finding, in addition to contributing a new benchmark,  is that current LLMs struggle significantly with active reasoning, performing far below perfect accuracy in these tasks.
The paper also shows that advanced prompting or training techniques yield only marginal improvements: methods like Tree-of-Thought (which searches through possible reasoning paths) and instruction fine-tuning or alignment (SFT, DPO) did not consistently or substantially boost accuracy. These results highlight that simply scaling up model size or applying existing reasoning tricks is not enough to reach reliable performance in active reasoning. The authors conclude that there is an urgent need for new techniques to improve LLMs’ ability to ask the right questions and use the answers effectively.
The paper’s contributions are thus: (1) introducing AR-Bench with well-defined tasks and evaluation metrics for active reasoning, (2) an empirical study revealing significant limitations of current LLMs in these scenarios, and (3) analysis tools (a “process score” metric based on key question completion) to quantify the quality of the model’s questioning strategy over time.

## update after rebuttal

No change in scores. I am already at 3 and I think I will maintain my score.

**Claims And Evidence:**

Support for Key Claims: The paper’s central claims are mostly well-supported by empirical evidence
- LLMs struggle at active reasoning: The experiments clearly demonstrate this. For example, GPT-4o achieves only 54% accuracy in Detective Cases, 63 F1 in Situation Puzzles, and 35% in Guessing Numbers​.  Smaller open models (LLaMA 8B/70B) fare even worse (often near zero on Guessing Numbers)​.
- Advanced methods give only marginal or inconsistent gains: For example, the Tree-of-Thought (ToT) method improved the 8B model’s accuracy on Detective Cases from 31% to 45%, but for the 70B model it decreased accuracy from 40% to 40% (no gain)​. Likewise, fine-tuning the model on the training puzzles (SFT) boosted detective case accuracy (e.g. LLaMA-8B from 32% to 62%) but completely failed on Guessing Numbers (0% correct). The direct preference optimization (DPO) alignment method had mixed results, even performing worse than zero-shot on some tasks​. The paper’s claim that these methods provide only “marginal improvements” is a bit modest – in some cases the improvements were sizable (e.g. +30 points for 8B SFT on Detective Cases​), but overall no method achieved high absolute performance or worked reliably across all tasks. Thus, the evidence supports the spirit of the claim: none of the tested state-of-the-art techniques is anywhere close to solving active reasoning.

### Questionable or Unsubstantiated Claims:
- Disparity with real-world tasks: The introduction asserts that passive reasoning alone is insufficient for many real-world problems and gives intuitive examples (travel planning requiring asking the client questions, medical diagnosis requiring patient Q&A). This claim is commonsensical and not really disputed, but the extent to which current models fail in realistic active scenarios isn’t directly proven in the paper since AR-Bench is composed of synthetic puzzles.

**Essential References Not Discussed:**

Related work section looks complete.

**Experimental Designs Or Analyses:**

- The authors evaluated a wide range of conditions (different models, multiple methods) on a reasonably large test set for each task (100 puzzles per task)​. This gives the results some statistical significance – they’re not anecdotal one-off examples, but averaged over 100 trials per task. They also use train/validation splits for fine-tuning, which helps avoid overfitting. The inclusion of multiple model sizes (8B vs 70B vs the extremely large 405B and GPT-4) allows analysis of scaling effects.
- A baseline to consider is optimal strategy (or human expert performance). The paper doesn’t provide any human evaluation or results from the Oracle policy. It would be illuminating to see what score a perfect questioner would get - presumably 100% on all tasks by design. Or at least, a reasonable heuristic strategy: e.g., for Guessing Numbers, there are known algorithms that could solve 100% within ~7 guesses. --For Detective Cases, a simple strategy could be to ask directly about each suspect’s motive, alibi, etc., which should solve it by elimination. The models did not approach anywhere near 100% on those tasks, highlighting the gap.
- The analysis doesn’t deeply break down errors beyond broad strokes. For a rigorous critique, one could say the paper could have included more error analysis: e.g., what types of questions did models tend to ask, where did they go wrong?

**Methods And Evaluation Criteria:**

- The three tasks in AR-Bench are well-chosen to cover a spectrum of reasoning with incomplete information. Each task explicitly requires the model to interact through questions/guesses rather than simply compute an answer from a given prompt, capturing the essence of active reasoning.
- As highlighted above, one possible concern is that these tasks, being synthetic, might not capture all nuances of real-world active reasoning. For instance, real information-seeking often involves unclear or unstructured answers, whereas AR-Bench’s NPC (non-player character) respondents give relatively clean, formatted feedback (yes/no, or narrative statements following a script). However, this is arguably a necessary simplification to start quantifying performance, and perhaps a step in the right direction. We would expect these benchmarks to get more realistic and difficult over time with more deceitful/multifaceted communication between the player and NPC.
- The authors created the dataset via a multi-step generation pipeline using GPT-4 (referred to as GPT-4o) to synthesize scenarios and ground-truth solutions​. This pipeline involves generating a core scenario, expanding it into a detailed story tree, extracting key facts as question-answerable bits, and then packaging the puzzle for the model​.  This is a clever approach to scale up data without manual writing.
- The evaluation is conducted as a multi-turn dialogue between the tested model (the “player”) and the environment (NPCs). They fixed the number of rounds (questions) to 25 for each session. . In most cases, if the model hasn’t solved the puzzle in 25 turns, it’s probably going in circles and indeed the paper’s analysis shows diminishing returns in late rounds.
- The evaluation metrics for final performance are straightforward and appropriate: Accuracy for Detective Cases (did you find the murderer or not), F1 score for Situation Puzzles (how well does the model’s final explanation overlap with the ground-truth explanation), and Exact Match for Guessing Numbers (did the model get the number exactly right)​. Accuracy and exact match are clear-cut. The use of F1 for the puzzles is reasonable since the solution is a free-form text describing the scenario; F1 (as commonly used in QA tasks) gives partial credit if the model’s answer includes some of the correct information​

**Other Comments Or Suggestions:**

-

**Other Strengths And Weaknesses:**

### Strengths:
- The paper does an excellent job of laying out a new benchmark with clarity. The tasks are well-motivated and the dataset is reasonably large considering the complexity of each example. The generation pipeline demonstrates originality in how to create diverse puzzles systematically. creating evaluation data for interactive reasoning is non-trivial – the authors provided a solution that future researchers can build on.
- The paper addresses a capability gap that is widely recognized as important for AI – the ability to interact and ask questions. In terms of impact, if the community adopts AR-Bench or is inspired by it, it could steer research toward more agentive AI, which is significant. The results are a bit surprising to those who assume GPT-4 is near human-level on all reasoning; highlighting this gap is an important contribution.
- Aside from the technical content, the paper is generally well-written. The objectives, setup, and findings are presented in a logical order. Key points are illustrated with figures and tables that are easy to read. The authors also took care to include any ethical considerations like fictionalizing violent scenarios.

### Weaknesses
- As discussed, the benchmark, while new, overlaps with prior and concurrent works. The paper could be perceived as incremental in that it packages known puzzle types and tests known models without proposing a new algorithm. Essentially, AR-Bench is a benchmark paper, and such papers are often judged by whether they’ll be widely used. The risk is that if the community decides to focus on a different benchmark, AR-Bench could be bypassed. Its success and relevance will depend on adoption.
- There have been few attempts to explain certain unexpected results, such as SFT's getting 0% accuracy on GN with a 70B model.

**Questions For Authors:**

1. Did you fine-tune one model on all three tasks jointly, or separate models per task? If jointly, could task interference have hurt the performance on the numeric task? Why do you think the fine-tuned models underperformed on some tasks (especially the 0% accuracy on Guessing Numbers)
2. Did you consider or attempt any of the active reasoning-specific prompting strategies from related work, such as “Proactive CoT” or “Uncertainty of Thoughts (UoT)”, on AR-Bench? It would be interesting to see their results on AR-bench given they specifically target this problem.
3. How do the NPCs handle odd or irrelevant questions from the player model? For example, if the model asks a nonsensical question or something not covered in the story, does the NPC respond with “I don’t know” or does it make something up? Handling such cases consistently can affect the fairness of evaluation – if an NPC accidentally provides a hint or gets confused, it could skew a round’s outcome. Any observations on the NPCs’ behavior would be helpful.
4. Do you think it is possible to characterize any model’s mistakes into categories; if they’re running into identical or closely related issues which stops them from reaching the correct answer? Like asking an unrelated question midway would take it off track and this could be a common error models across scale would commit.

**Relation To Broader Scientific Literature:**

This work fits into a quickly growing body of literature on enabling and evaluating interactive reasoning in language models. The authors do acknowledge many related works:
- Clarification Question Benchmarks: For example, Qulac (Aliannejadi et al., 2019) is a dataset of ambiguous search queries where models propose a question to clarify user intent​. Abg-CoQA (Guo et al., 2021) similarly deals with ambiguous questions in a conversational QA setting​. These works established the importance of asking the right question when faced with ambiguity, but they typically involve just one question-response exchange before answering. AR-Bench goes beyond by requiring multiple rounds of questioning and complex inference, which is a step up in difficulty.
- Twenty Questions and Guessing Games: Perhaps the closest precursors to AR-Bench are benchmarks inspired by the classic 20 Questions game. Abdulhai et al. (2023) introduced LMRL-Gym, a suite of language-game environments for reinforcement learning, which included a 20 Questions game and a “Guess my City” game​. In these, the model must identify a hidden topic or city by asking yes/no questions, very much like AR-Bench’s Situation Puzzles or a constrained version of Guessing Numbers. In fact, AR-Bench’s Guessing Numbers (GN) task can be seen as a variant of a guessing game, but with numeric feedback instead of simple yes/no. The novelty of AR-Bench’s GN is slight in content (number guessing has been around for decades as a game), but novel in being used to evaluate LLMs systematically. It provides a symbolic reasoning test that wasn’t explicitly in LMRL-Gym (which focused on yes/no only). On the other hand, the 20 Questions task from LMRL-Gym covers similar grounds of information-seeking.
- Other concurrent works have targeted active reasoning in specialized contexts. MediQ (Li et al., 2024c) is a benchmark for interactive medical diagnosis dialogues​. In MediQ, an LLM plays doctor, asking a patient (simulated) questions to gather symptoms and test results, before making a diagnosis​. This is clearly a real-world analog of active reasoning. Similarly, a Troubleshooting environment (mentioned as Hu et al., 2024 in the paper) tackles technical problem solving through Q&A​. These domain-specific benchmarks share the same spirit as AR-Bench: the model must proactively query information. The difference is the domain knowledge required – MediQ expects medical knowledge, troubleshooting expects understanding of devices – whereas AR-Bench’s tasks are mostly domain-general or self-contained fiction.
- The novelty is incremental rather than completely groundbreaking. Each individual task has antecedents (e.g., bulls and cows game has existed since the 19th century; it’s novel only that we test an LLM on it). The scientific contribution lies in the empirical findings about LLM capabilities and the thoroughness of the benchmark. Those findings – that even GPT-4 struggles with these puzzles – might not shock those who have informally played such games with ChatGPT.

**Theoretical Claims:**

This paper is largely empirical and does not present new theoretical formalisms or proofs. Most claims are qualitative or quantitative observations about model performance. Therefore, there aren’t traditional “theorems” or mathematical proofs to verify in the work. The authors’ arguments are logical rather than formally theoretical.

---

> ### Author Rebuttal · Authors · 2025-04-01
>
> Thanks for the valuable feedback. To solve these concerns and questions, we provide point-to-point responses as follows.
>
> > Q1. The benchmark’s novelty and consistency with real-world scenarios.
>
> **Reply:** We answer this question from three aspects.
>
> **(1) Reality.** Real-world active reasoning can be messy, but benchmarks initially need simpler, controlled tasks to isolate specific capabilities, i.e., effectively asking questions to obtain missing information. Synthetic puzzles ensure consistency and clear metrics, while the yes/no feedback is more structured than real interactions. We plan to add more nuanced “deceitful” NPCs over time.
>
> **(2) Novelty.** Although our puzzles resemble known games, combining them into a single benchmark systematically tests active reasoning, which is an overlooked dimension. We provide 6,000+ puzzles, curated prompts, automated feedback, and extensive experiments, showing current LLMs struggle primarily due to inadequate questioning. Pinpointing *how* and *why* it fails is our key contribution.
>
> **(3) Influence.** We will release all data and code publicly, enabling easy reproduction and further improvement. AR-Bench fills a gap in current reasoning benchmarks, which largely assume complete information is available. Our controlled environment highlights specific failings and can evolve to include richer, more realistic forms of active reasoning.
>
> > Q2. The error patterns of LLMs on AR-Bench.
>
> **Reply:** We carefully check the 600 running dialogs of Llama3.1-8B and GPT-4o on AR-Bench and categorize typical errors in each task.
>
> In DC:
>
> - Timeline Misinterpretation (TM): Models confuse the timeline of the murderer and ask for information of an irrelevant time.
> - Evidence Overlooked (EO): Models ask for details and overlook the key evidence needed to find the murderer.
>
> In SP:
>
> - Evidence Overlooked (EO): Models jump to conclusions without uncovering supporting evidence.
> - Unsupported Assumptions (UA): Models fabricate fake details in the conclusion.
>
> In GN:
>
> - Feedback Misunderstanding (FM): Models misinterpret feedback, failing to keep correct digits or eliminate wrong ones.
> - Incomplete Testing (IT): After identifying a correct digit, models fail to determine its correct position.
>
> The following tables show error statistics for Llama3.1-8B and GPT-4o.
>
> - In DC, GPT-4o overlooks less key evidence but struggles more with timeline issues.
> - In SP, Llama3.1-8B fabricates details more often, while GPT-4o tends to miss details.
> - In GN, Llama3.1-8B makes more errors overall, though GPT-4o also has high error rates.
>
> |Task|Error Type|Llama3.1-8B|GPT-4o|
> |:-|:-|:-|:-|
> |DC|TM|10%|31%|
> ||EO|61%|15%|
> |SP|EO|36%|44%|
> ||UA|90%|72%|
> |GN|FM|78%|61%|
> ||IT|81%|55%|
>
> > Q3. Explain the technical details and empirical results of SFT.
>
> **Reply:** We train the model separately for each task. Our analysis of the SFT results reveals the following:
>
> **SFT tends to memorize training data rather than develop active reasoning skills.** This is especially problematic in the GN task, where memorizing number patterns doesn’t teach the logical deduction needed for accurate guessing, resulting in 0% accuracy on unseen test cases.
>
> **Moreover, the training data includes only the output questions, without showing the underlying reasoning behind them.** This lack of reasoning traces limits learning, especially in symbolic tasks like GN. In contrast, DC data (with its free-form questions) better supports the development of active reasoning, leading to improved performance.
>
> > Q4. Attempt related work Proactive CoT and UoT on AR-Bench.
>
> **Reply:** We adapt these two methods to AR-Bench:
>
> - **Proactive CoT**: Pre-defined task-specific strategies and actions based on human reasoning patterns.
> - **UoT**: In DC, we sample 3 question branches, simulate 1 turn, and estimate uncertainty reduction. In SP, we initialize the potential answers, sample 3 questions/turn, and estimate uncertainty reduction. In GN, we sample 3 guess branches, simulate 1 turn, and use the model to estimate uncertainty reduction.
>
> The results below show that both methods cannot solve AR-Bench well. This demonstrates the necessity of designing new methods (please see Q3 to reviewer Briw).
>
> |Method|Base Model|Proactive CoT|UoT|
> |:-|:-|:-|:-|
> |DC|31|31|10|
> |SP|43|47|33|
> |GN|0|0|1|
>
> > Q5. The robustness of the LLM judges against irrelevant questions.
>
> **Reply:** We instruct judge models to withhold useful information when questions are irrelevant. In SP, they respond with “unknown”. In CD, while suspects may reply, they avoid revealing useful information. We illustrate this with [running examples](https://anonymous.4open.science/r/AR-Rebuttal-BC37/1zoX/Q5/examples.png) of irrelevant questions.
>
> > Q6. Human evaluation on AR-Bench.
>
> **Reply:** Due to the high cost of large-scale human evaluation, we conducted a small demo with two undergraduate students on AR-Bench to show human performance. Please see our response to Q4 for Reviewer Vrs4.

---

### Official Review · Reviewer_Vrs4 · 2025-03-13

**Overall Recommendation:** 2

**Summary:**

The paper addresses a significant research gap in evaluating Large Language Models (LLMs) for active reasoning, where models must actively query external sources due to incomplete information, rather than passively reasoning from complete data. The authors propose AR-Bench, featuring three active reasoning tasks: detective cases, situation puzzles, and guessing numbers. They show that modern LLMs, including state-of-the-art models like GPT-4o, perform poorly in these tasks, highlighting a major gap between passive and active reasoning abilities. The authors demonstrate that even sophisticated reasoning techniques such as tree-of-thought (ToT) and post-training optimization methods like supervised fine-tuning (SFT) and direct preference optimization (DPO) yield only marginal improvements.

**Claims And Evidence:**

**Claim:** LLMs struggle to consistently generate high-quality, effective questions during multi-turn interactions.

**Evidence**: Quantitative results from experiments conducted on AR-Bench demonstrate low accuracy scores for LLMs, such as GPT-4o achieving only 35% accuracy on the guessing numbers task. Qualitative case studies highlighting examples of vague or ineffective questions posed by models during experiments, reinforcing observed limitations

**Claim:**Inference-time optimizations, such as increasing interaction rounds, have minimal effects in improving LLMs' active reasoning performance.

**Evidence**:Comparative analysis of models (e.g., Llama-3.1-8B vs. GPT-4o) across tasks provides empirical evidence showing marginal improvements with advanced techniques.

**Essential References Not Discussed:**

Yes.

**Experimental Designs Or Analyses:**

* Evaluations include various LLMs (Llama-3.1-8B, 70B, 405B, GPT-4o-mini, GPT-4o).
* Reasoning methods evaluated include zero-shot, few-shot, few-shot with instructions, tree-of-thought (ToT), supervised fine-tuning (SFT), and direct preference optimization (DPO).
* Performance metrics include accuracy (detective cases), F1-score (situation puzzles), and exact match rate (guessing numbers).
* Multi-round evaluation measures the progression in active reasoning capabilities over interactions.

**Methods And Evaluation Criteria:**

* The authors introduce AR-Bench, which contains three tasks (detective cases, situation puzzles, guessing numbers) to evaluate the active reasoning of LLMs.

* Active reasoning capabilities are assessed by simulating multi-turn interactions where the model must iteratively formulate questions or guesses and interpret external feedback.

* Performance metrics are accuracy (detective cases), F1-score (situation puzzles), and exact match (guessing numbers).

* Tasks effectively test different reasoning types—commonsense, logical, symbolic—providing comprehensive coverage

* The models considered for the evaluation for this particular task doesn't really makes sense as the GPT4o and GPt-4o mini are reasoning models as mentioned by OpenAI and the comparison of these models with zero-shot performance of LLama models, doesn't really make sense.

* Also the benchmark created is a synthetic data, there is no human intervention in any part, I feel the synthetic data has to be evaluated for it's accuracy as well, if the data created itself its wrong then it's no use.

* Then the models selected for this task are very limited, just two families of models, the authors should have considered more wide range of models with more diverse architectures, instead of considering one open source and one closed course and comparing them. Also the methods considered were only evaluated on two models of same family, which doesn't really give much of insights.

* Also the details of the model configurations are not mentioned, maybe they should mention the temperatures and topk etc, try different values to see how the creativity differs, as the tasks needs more reasoning.  High temperatures are generally used for creative tasks.

**Other Comments Or Suggestions:**

N/A

**Other Strengths And Weaknesses:**

## Strengths

* The paper addresses an important yet underexplored aspect of large language models: active reasoning in contexts with incomplete information.
* The paper is very well written and the details are very clear. Clear distinction between passive and active reasoning is well explained, accompanied by illustrative figures, making the work easy to comprehend.

## Weaknesses

* Absence of direct human evaluation in assessing question quality or reasoning effectiveness limits verification of the benchmark’s real-world relevance and reliability.
* Others mentioned in the above Methods section.

**Questions For Authors:**

N/A

**Relation To Broader Scientific Literature:**

It's a good benchmark, but needs more robustness.

**Theoretical Claims:**

N/a

---

> ### Author Rebuttal · Authors · 2025-04-01
>
> Thanks for the valuable feedback. To solve these concerns and questions, we provide point-to-point responses as follows.
>
> > Q1. The selection of language models to compare.
>
> **Reply:** We provide a two-fold answer as follows.
>
> **(1) Considering the huge expense of computing and API calls, we evaluate the currently representative LLMs on AR-Bench.** Specifically, we select GPT-4o and GPT-4o-mini as representatives of closed-source LLMs; and Llama3.1 8B, 70B, and 405B as representatives of open-source LLMs. These models are widely used and compared in current leaderboards and arenas, which include both reasoning and non-reasoning models. In our experiments, we guarantee fair evaluations and comparisons by providing the same information to different models under evaluation.
>
> **(2) We conduct extra experiments with more language models and show their general deficiency in active reasoning tasks.** We involve Qwen2.5 (3B, 7B) - another representative model family, and QwQ-32B - a powerful reasoning model. The evaluation result shown below is consistent with our observation. This highlights the widespread challenges that current LLMs face in active reasoning tasks.
>
> |Model|DC|SP|GN|
> |:-|:-|:-|:-|
> |GPT-4o-mini|62|41|6|
> |GPT-4o|54|63|35|
> |Llama3.1-8B|31|43|0|
> |Llama3.1-70B|55|54|8|
> |Llama3.1-405B|43|55|12|
> |Qwen2.5-3B|19|34|0|
> |Qwen2.5-7B|12|51|4|
> |QwQ-32B|57|49|1|
>
> > Q2. Ablation study with various model configurations of temperatures and topk.
>
> **Reply:** In our experiments, we used consistent hyperparameter settings with temperature = 0.7 and top-p = 0.7. We did not modify the top-k parameter, as it is not configurable in the OpenAI API. **Further, we present extra ablation studies and show that varying temperature and top-p values resulted in only marginal differences in performance across active reasoning tasks**. As shown in the tables below, extreme values of temperature and top-p (either too high or too low) lead to suboptimal performance in SP and DC tasks, while having minimal impact on GN.
>
> |temperature|0|0.3|0.5|0.7|1|
> |:-|:-|:-|:-|:-|:-|
> |DC|39|35|39|31|37|
> |SP|35|34|35|43|31|
> |GN|2|0|0|0|2|
>
> |top-p|0|0.3|0.5|0.7|1|
> |:-|:-|:-|:-|:-|:-|
> |DC|32|40|45|31|44|
> |SP|32|36|37|43|34|
> |GN|1|0|1|0|0|
>
> > Q3. The human intervention in generating the dataset.
>
> **Reply:** We answer this question in three folds:
>
> **(1) We used a rigorous multi-stage process to ensure the reliability and logical consistency of each AR-Bench sample.** Specifically:
>
> - For DC, the core logic remains intact. Each puzzle has a uniquely identifiable murderer based on motive, opportunity, and weapon access. The synthetic process only adds narrative details without affecting the deductive structure.
> - For SP, which is based on lateral thinking, any logically consistent explanation is valid. We start by sampling a core counterfactual fact and use tree-based expansion to generate explanatory questions and corresponding facts (see Figure 4).
> - For GN, the task is simple and the synthetic process is reliable, as it only involves generating four-digit numbers with unique digits.
>
> **(2) We conducted manual checking of the puzzles and ground truths in AR-Bench to ensure they are reliable, logically consistent, and solvable through reasoning.** Specifically:
>
> - For DC, we verify that the murderer has all key traits (motive, opportunity, weapon access) and that innocent suspects lack at least one.
> - For SP, we check that the provided answer logically explains the scenario and maintains internal consistency.
> - For GN, we confirm that each number consists of four unique digits.
>
> Besides, as introduced in Appendix A, we provided the source files of AR-Bench in an [anonymous repository](https://anonymous.4open.science/r/AR-Bench-submission-code-591C). The reviewers can directly check the quality of data points in the AR-Bench.
>
> **(3) In addition, synthetic data offers several advantages.** Synthetic data reduces overlap with existing datasets for mitigating data leakage, scales efficiently without the cost of human-annotated examples, and is easily adaptable for creating new active reasoning tasks.
>
> > Q4. The human evaluation of the benchmark.
>
> **Reply:** Considering the unaffordable expense of large-scale human evaluation, we conduct an extra demo-level human evaluation with two undergraduate students on AR-Bench to show human performance (please see the table below). Despite the inherent challenges in human evaluation, the results still reveal a substantial performance gap between humans and LLMs, underscoring the critical need to enhance active reasoning capabilities in current language models.
>
> ||DC|SP|GN|
> |:-|:-|:-|:-|
> |Human Evaluation|80|67|100|
> |GPT-4o|54|63|35|
>
> We will include the above clarifications and discussions in the revision. Any further comments and discussions are welcome! Thank you!

---

### Decision · Program_Chairs · 2025-05-01

**Decision:**

Accept (poster)

**Comment:**

This paper presents a clear, well-structured benchmark (AR-Bench) that addresses a timely capability gap in LLMs: active reasoning via interactive question-asking. While the benchmark is synthetic and the paper does not propose a novel method, it systematically exposes current LLMs' limitations, analyzes common failure modes, and introduces a testbed that could drive future work in agentic reasoning. The benchmark is well-motivated, empirically solid, and thoughtfully executed. I recommend acceptance.